# Structural visualization of small molecule recognition by CXCR3 uncovers dual-agonism in the CXCR3-CXCR7 system

Shirsha Saha[1,4], Fumiya K. Sano[2,4], Saloni Sharma[1,4], Manisankar Ganguly[1], Annu Dalal[1], Sudha Mishra[1], Divyanshu Tiwari[1], Hiroaki Akasaka[2], Takaaki A. Kobayashi[2], Nabarun Roy[1], Nashrah Zaidi[1], Yuzuru Itoh[2], Rob Leurs[3], Ramanuj Banerjee[1] ✉, Wataru Shihoya[2] ✉, Osamu Nureki[2] ✉ & Arun K. Shukla[1] ✉

Chemokine receptors are critically involved in multiple physiological and pathophysiological processes related to immune response mechanisms. Most chemokine receptors are prototypical GPCRs although some also exhibit naturally-encoded signaling-bias toward β-arrestins (βarrs). C-X-C type chemokine receptors, namely CXCR3 and CXCR7, constitute a pair wherein the former is a prototypical GPCR while the latter exhibits selective coupling to βarrs despite sharing a common natural agonist: CXCL11. Moreover, CXCR3 and CXCR7 also recognize small molecule agonists suggesting a modular orthosteric ligand binding pocket. Here, we determine cryo-EM structures of CXCR3 in an Apo-state and in complex with small molecule agonists biased toward G-proteins or βarrs. These structural snapshots uncover an allosteric network bridging the ligand-binding pocket to intracellular side, driving the transducer-coupling bias at this receptor. Furthermore, structural topology of the orthosteric binding pocket also allows us to discover and validate that selected small molecule agonists of CXCR3 display robust agonism at CXCR7. Collectively, our study offers molecular insights into signaling-bias and dual agonism in the CXCR3-CXCR7 system with therapeutic implications.

Chemokines are small proteins secreted by immune cells, that play critical roles in a myriad of physiological processes including cellular migration and inflammatory responses, by activating chemokine receptors[1,2]. Chemokine receptors belong to the superfamily of G-protein-coupled receptors (GPCRs) with primary coupling to the Gαi subtype of heterotrimeric G-proteins and β-arrestins (βarrs)[3,4]. They are expressed on a variety of immune cells with wide-ranging contributions to various aspects of our immune response mechanisms, and their aberrant signaling is implicated in multiple disease conditions including cancer[5,6], allergy[7,8], psoriasis[9], atherosclerosis[10], and autoimmune disorders[11,12]. While chemokine receptors exhibit a conserved seven-transmembrane architecture characteristic of prototypical GPCRs, there exists a significant promiscuity amongst chemokines for the chemokine receptors[13], making them an interesting system to explore the principles of ligand-directed signaling bias. Moreover, several chemokine receptors, referred to as the atypical chemokine receptors (ACKRs) or Arrestin-Coupled Receptors (ACRs), exhibit naturally encoded intrinsic-bias, as they do not exhibit any

[1]Department of Biological Sciences and Bioengineering, Indian Institute of Technology Kanpur, Kanpur, India. [2]Department of Biological Sciences, Graduate School of Science, The University of Tokyo, Tokyo, Japan. [3]Amsterdam Institute for Molecules, Medicines, and Systems (AIMMS), Division of Medicinal Chemistry, Faculty of Sciences, VU University Amsterdam, Amsterdam, The Netherlands. [4]These authors contributed equally: Shirsha Saha, Fumiya K. Sano, Saloni Sharma. ✉e-mail: ramanujb@iitk.ac.in; wtrshh9@gmail.com; nureki@bs.s.u-tokyo.ac.jp; arshukla@iitk.ac.in

functional coupling to G-proteins while maintaining robust interaction and signaling through βarrs upon activation by their natural chemokine agonists[14,15]. In this context, CXCR3 and CXCR7 constitute an interesting receptor pair, wherein CXCR3 couples to both G-proteins and βarrs while CXCR7 lacks any measurable G-protein activation but robustly recruits βarrs upon agonist-stimulation[16–18].

CXCR3 and CXCR7 are expressed on a variety of immune cells such as innate lymphocytes, effector T cells, plasmacytoid dendritic cells, subsets of B cells, and also within the tumor microenvironment[19–26]. Aberrant expression and signaling of CXCR3/CXCR7 are implicated in glomerulonephritis, inflammatory and neuroinflammatory disorders, rheumatoid arthritis, atherosclerosis, heart failure, pulmonary fibrosis, multiple sclerosis, and autoimmune disorders, making them important therapeutic targets[27–35]. While CXCR3 is activated by three different C-X-C type chemokines, namely CXCL9, 10, and 11, CXCR7 recognizes CXCL11 and CXCL12[36]. Interestingly, CXCL11 appears to act as a βarr-biased agonist at CXCR3, as compared to CXCL9 and CXCL10[37–40], and also promotes the formation of non-canonical CXCR3-Gαi-βarr complexes, as demonstrated elegantly in cellular context[41]. Moreover, a splice variant of CXCR3, referred to as CXCR3-B, contains an extended N-terminal domain and exhibits a differential transducer-coupling profile and signaling bias as compared to the CXCR3-A splice variant[42–45]. CXCR3 and CXCR7 are also among the very few examples of chemokine receptors that are activated by small molecule agonists[46–48]. For example, VUF11418 and VUF10661 have been developed as G-protein-biased and βarr-biased agonists of CXCR3[49], respectively, while VUF11207 is characterized as a CXCR7 selective agonist in terms of eliciting βarr recruitment and activation[18]. It is worth noting that VUF11418 and VUF10661 are relatively biased agonists at CXCR3, i.e., they both activate the G-protein pathway to comparable levels and the bias results from their differential efficacy in inducing βarr recruitment[49]. It is also noteworthy that CXCR7 was initially considered to be a decoy receptor with a primary function of regulating the chemokine gradient in circulation and tissue microenvironments. However, recent studies have not only established activation-dependent βarr recruitment but also provided clear hints of downstream signaling leading to cellular migration and invasion[17,18,50].

In this work, we elucidate the structural basis of activation and signaling bias at CXCR3 upon binding small molecule agonists and, guided by structural snapshots, discover the dual-agonism of these agonists at CXCR3 and CXCR7. We also uncover a potential allosteric network of local conformational changes and intramolecular interaction networks driving transducer-coupling bias at these receptors. Taken together, our study provides insights into activation, dual-agonism, and biased signaling at CXCR3 and CXCR7 with possible implications for the development of therapeutic molecules.

## Results

### Overall structures of CXCR3-G-protein complexes
In order to determine the structures of CXCR3, we purified recombinant CXCR3 in the presence of VUF11418 and VUF10661 (Fig. 1a, b), and subsequently, reconstituted their complexes with heterotrimeric G-proteins, stabilized using scFv16, as carried out previously for other GPCRs[51–53]. For complex reconstitution, we used miniGαo protein, an engineered variant of G-protein which comprises only the GTPase domain of the heterotrimeric G-protein α subunit, owing to its high stability[54,55]. We determined the structures of both complexes using cryo-EM at an approximate resolution of 3 Å (Fig. 1c, d and Supplementary Figs. 1–4). In addition, we also purified CXCR3 in the presence of CXCL10 and reconstituted its complex with G-protein followed by cryo-EM structure determination at 3.3 Å. However, the density for CXCL10 was not observed and therefore, this structure is referred to as the Apo-state (Fig. 1e). The precise details of the different components of these complexes modeled in the presented structures are summarized in Supplementary Fig. 5. In these cryo-EM structures, all the components of the complexes were reasonably well resolved including the small molecule agonists in the orthosteric binding pocket. However, some of the regions of the receptor such as the ECL1, ECL2, and extracellular portions of TM1-3 in the Apo-state, and the extracellular region of TM1 in the VUF11418-bound structure were not resolved well. A detailed list of the resolved residues for each component is provided in Supplementary Fig. 6. Expectedly, CXCR3 adopts a seven-transmembrane architecture with overall features resembling a typical active conformation, as observed for other class A GPCRs (Fig. 1), and the three structures exhibit an overall RMSD of <1 Å.

### Agonist-receptor interactions in the orthosteric binding pocket
The ligand binding pocket in CXCR3 is covered by ECL2 at the extracellular surface, which adopts a β-hairpin conformation encompassing residues Ser191[ECL2] to Tyr205[ECL2] (Supplementary Fig. 7a, b). The bulky VUF10661 adopts an "inverted U" shaped binding pose and exhibits a shallower binding mode, as opposed to VUF11418, which penetrates deeper into the orthosteric pocket of the receptor adopting a "linear" conformation (Fig. 2a, b). VUF10661 and VUF11418 occupy a position at a vertical distance of ~4.3 Å and ~3.8 Å, respectively, as measured from the conserved "toggle switch" residue Trp268[6.48] (Fig. 2a, b) and an overall similar orthosteric pocket (Fig. 2c). Interestingly, the ligand binding site in CXCR3 is encapsulated by a cluster of aromatic residues that we refer to as an "aromatic cage" (Fig. 2a, b), and it involves, for example, Tyr60[1.39], Trp109[2.60], Phe131[3.32], Phe135[3.36], Tyr205[ECL2], Trp268[6.48], Tyr271[6.51] and Tyr308[7.43] in the VUF10661-bound structure (Fig. 2d). A number of residues from TM1-3, TM5-7 and ECL2 help stabilize the "inverted U" shaped conformation of VUF10661 within the ligand binding pocket (Fig. 2d and Supplementary Data 1). For example, the hydroxyl group of Tyr60[1.39] makes hydrogen bonds and ionic interactions with the O12 and N3 of the ligand respectively, while Trp109[2.60] engages in π–π interaction with the C17−C22 ring. Likewise, Gln219[5.42] and Tyr308[7.43] pack against the C37−C42 ring of the ligand through anion–π interactions. A network of non-bonded contacts further stabilizes VUF10661 in the orthosteric pocket. While Tyr308[7.43] interacts with C35, Arg216[5.39] engages with C44/C45, and Tyr271[6.51] contacts C36/C38 of the ligand.

In contrast with VUF10661, VUF11418 adopts a linear planar conformation, and it is oriented at an angle of 45° to the plane of the lipid bilayer (Supplementary Fig. 7c). However, similar to VUF10661, VUF11418 also penetrates deep into the orthosteric pocket, and positions itself in an aromatic microenvironment surrounded by residues Trp109[2.60], Phe131[3.32], Phe135[3.36], Trp268[6.48], Tyr271[6.51] and Tyr308[7.43] (Fig. 2b). In addition, the toggle switch residue Trp268[6.48] undergoes a rotation of ~60° away from the ligand binding pocket, and several residues from TM2, TM3, TM6, and TM7 help stabilize VUF11418 interaction with the receptor (Fig. 2e and Supplementary Data 1). The side chain hydroxyl group of Ser304[7.39] forms an anion–π interaction with the C14−C19 ring, while Phe135[3.36] is involved in π–π interaction with the C20−C25 ring of the ligand. Several other non-bonded interactions help to further stabilize the ligand in the extracellular pocket of the receptor including the engagement of Leu102[2.53] and Phe131[3.32] with the iodine of the ligand while Trp268[6.48] packs with C23 of the C20−C25 ring. Furthermore, Tyr271[6.51] makes contacts with C11, C15, C16, C21 and C22 of VUF11418 (Supplementary Data 1).

Interestingly superimposition of the Apo-CXCR3 structure with the VUF10661/11418-bound CXCR3 structures reveals an RMSD of <1 Å across the main-chain Cα atoms. A comparison of the Apo-state CXCR3 structure with the inactive state and VUF-bound structures reveals that Phe182[4.56] in the Apo-state undergoes a rotation of 135° and points toward the ligand binding pocket, and this rotameric conformation is stabilized by Asn132[3.33] and Gln219[5.42] (Fig. 2f). In addition, Gln219[5.42] makes an ionic contact with Tyr271[6.51] and further locks the receptor in the active-like conformation. However, it is not possible to

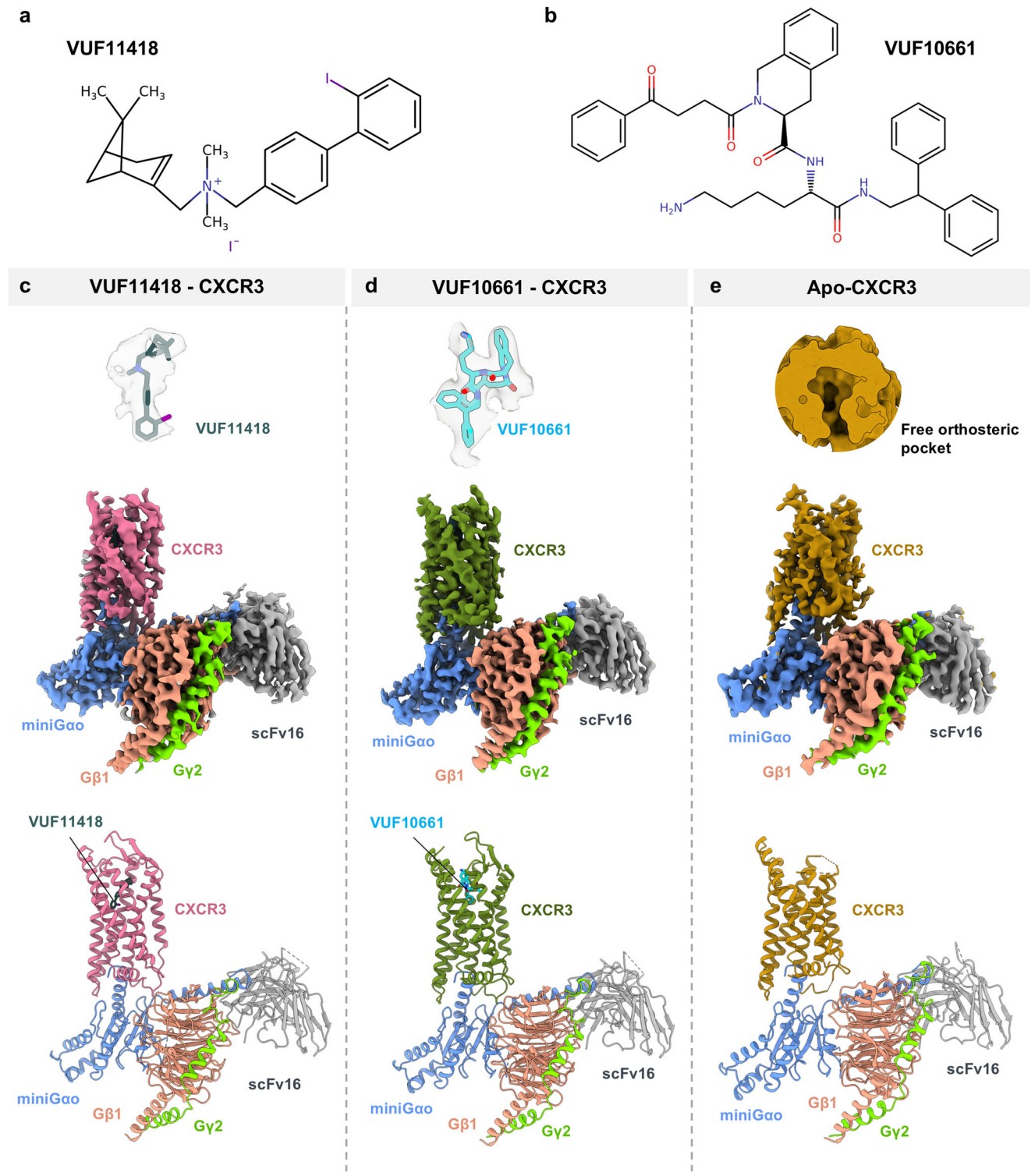

**Fig. 1 | Overall architecture of VUF11418-bound, VUF10661-bound, and Apo-CXCR3-G-protein complexes. a, b** Chemical structure of VUF11418 (**a**) and VUF10661 (**b**). Chemical structures have been prepared using Marvin JS. **c**–**e** Map and ribbon diagram of the ligand-bound and Apo-CXCR3-Go complexes (front view) and the cryo-EM densities of the ligands (sticks) are depicted as transparent surface representations. (VUF11418-CXCR3: pale violet red, VUF11418: deep teal, VUF10661-CXCR3: olive drab, VUF10661: cyan, Apo-CXCR3: dark goldenrod, mini-Gαo: cornflower blue, Gβ1: light coral, Gγ2: chartreuse, scFv16: gray).

discern at this point whether these features represent receptor activation due to constitutive activity or reflect the stabilization of an active conformation by CXCL10, which has subsequently dissociated from the receptor.

Taking lead from our structures, we carried out an extensive mutagenesis analysis wherein we mutated the ligand interacting residues on CXCR3 individually to alanine and measured the effect of the same on G-protein signaling and βarr recruitment following stimulation with VUF10661 and VUF11418 (Fig. 3a, b). We observed an excellent correlation between the structural observations and the effect of mutating the corresponding residues on agonist-induced functional responses. Interestingly, we found that one of the

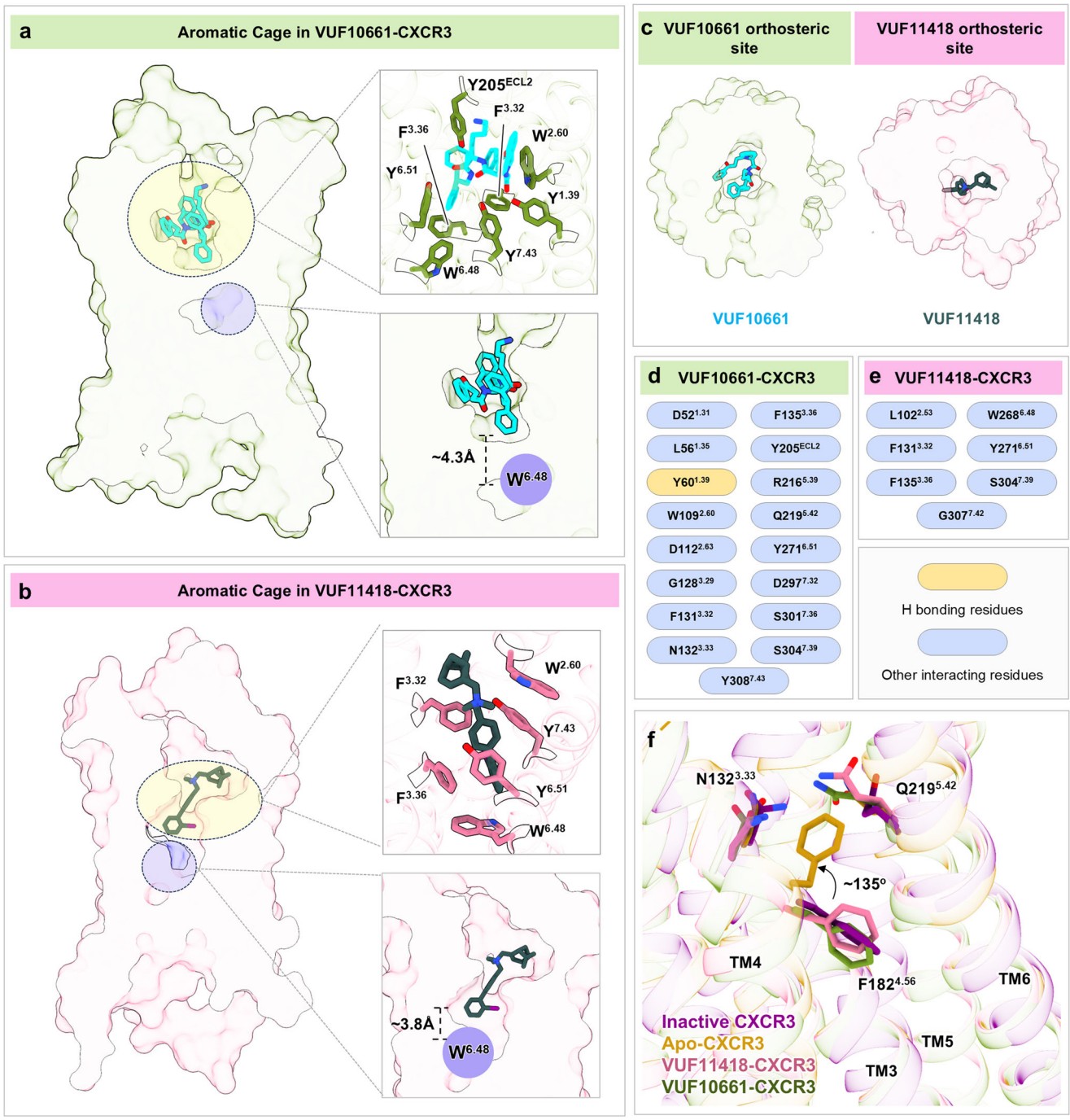

**Fig. 2 | Ligand binding interface and aromatic cage in CXCR3. a, b** Cross-section of the orthosteric pocket in CXCR3 depicting the aromatic cage and the depth of penetration of the ligands with respect to the conserved W[6.48]. **c** Orthosteric pocket in CXCR3 showing bound VUF11418 and VUF10661 in inverted 'U' and linear shaped conformations, respectively. **d, e** List of CXCR3 residues interacting with the ligands VUF11418 and VUF10661. **f** Rotameric transitions in Apo-CXCR3 when compared with the inactive state and VUF-bound CXCR3 structures. (AMG487-CXCR3: purple; PDB: 8K2W, VUF11418-CXCR3: pale violet red, VUF11418: deep teal, VUF10661-CXCR3: olive drab, VUF10661: cyan, Apo-CXCR3: dark goldenrod).

mutants, CXCR3[W109A] exhibited a near-complete loss of both G-protein activation and βarr recruitment upon stimulation with VUF10661 but exhibited a slightly enhanced transducer-coupling profile compared to wild-type receptor when stimulated with VUF11418 (Figs. 3a, b and 4a). While W109[2.60] directly interacts with VUF10661, it undergoes an outward angular shift of ~87° away from the orthosteric pocket in the VUF11418-bound structure (Fig. 4b). Therefore, it is possible that mutation of this bulky residue to alanine induces a void, which would allow the extracellular portion of TM2 to move toward the orthosteric pocket, allowing additional interactions

and stabilization of VUF11418 in the orthosteric pocket of CXCR3 leading to a measurable enhancement in the functional response. Furthermore, mutating Y271[6.51] to alanine results in some reduction of G-protein activation but near-complete loss of βarr recruitment in response to both VUF10661 and VUF11418 (Figs. 3a, b, and 5a). Structural analysis indicates that Y271[6.51] interacts with both the VUFs (Fig. 5b) and mutation of this residue to alanine would lead to the generation of a minor sub-pocket adjacent to the VUF ligand binding site, which might in turn allow space for an inward movement of the TM6 toward the orthosteric pocket of the receptor. This may

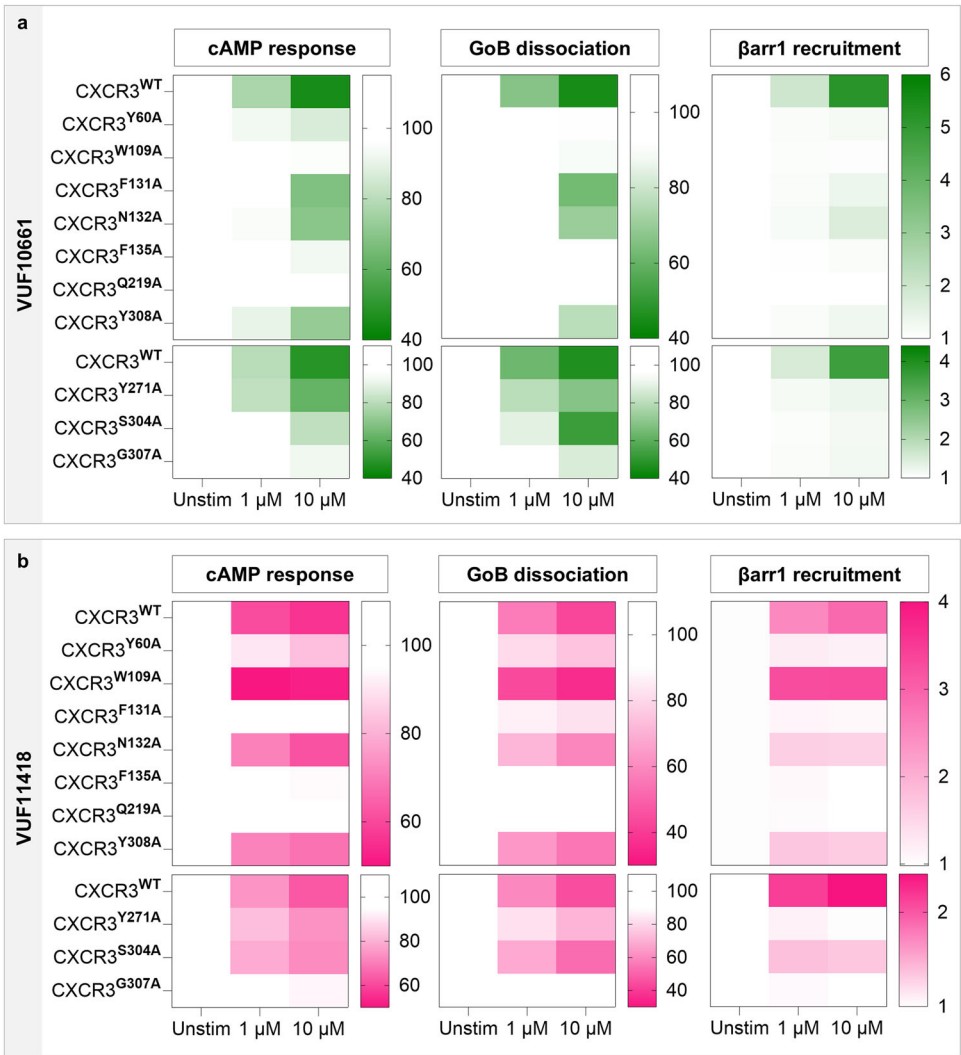

**Fig. 3 | Effect of mutating CXCR3 ligand interacting residues on transducer coupling. a, b** Heatmap showing G-protein signaling, as measured using cAMP secondary messenger response (Gi-mediated decrease in cytosolic cAMP levels) and GoB dissociation (indicated by a decrease in luminescence), and βarr1 recruitment (indicated by an increase in luminescence) downstream to CXCR3 mutants following stimulation with VUF10661 (green) and VUF11418 (pink). Data (mean ± SEM) represents either three (for G-protein assays) or four (for βarr1 recruitment) independent experiments, performed in duplicate, and has been normalized with respect to the signal observed in unstimulated condition, treated as either 100% (for G-protein assays) or 1 (for βarr1 recruitment). Source data are provided as a Source Data file.

stabilize a CXCR3 conformation that drives signaling through G-proteins. However, further structural exploration of the conformational changes in CXCR3 upon Y271$^{6.51A}$ mutation should allow corroboration of this hypothesis in future studies.

**Comparative analysis of CXCR3 structures**

Structural superimposition of the VUF11418-bound CXCR3 structure determined here and the VUF11222-bound CXCR3 structure reported recently[56] (PDB: 8HNM) reveals an overall similar receptor conformation and ligand interaction mode in the orthosteric binding pocket (Fig. 6a). Both these ligands penetrate to a similar depth and occupy a similar position within the ligand binding pocket with the biaryl group facing downward in a spatially aligned pose. However, we also observe some modest but interesting differences between the binding of these two ligands to the receptor. For example, the bicycloaliphatic group in the two ligands displays an opposite orientation along the vertical axis of the receptor (Fig. 6b). This is also reflected in significant rotameric rearrangements of specific residues in the receptor such as Trp109$^{2.60}$ and others in the aromatic cage to accommodate the corresponding ligands (Fig. 6c, d). Additionally, we also observe a conformational shift

in ICL1 and ICL2; however, the same could be arising as a result of the different G-proteins used in the two studies (Fig. 6e, f).

Similarly, structural superimposition of VUF10661-bound CXCR3 structure determined here and PS372424-bound CXCR3 structure reported recently[56] (PDB: 8HNL) also reveals an overall similar receptor conformation and ligand interaction mode in the orthosteric binding pocket (Fig. 6g). Similar to VUF10661, PS372424 also adopts an "inverted U"-shaped conformation in the orthosteric binding pocket of the receptor, spatially aligns with the former, and shares similar interacting residues on CXCR3 (Fig. 6h, i). Despite the similarities, there are some notable differences as well, which include additional contacts of the diphenylethylamine group in VUF10661 with the receptor and differences in the positioning of the ICL2 and ICL3 (Fig. 6j–m). We also note that some of these differences, especially those on the cytoplasmic side of the receptor may reflect the type of G-protein used in the two studies, i.e., miniGαo in the current study vs the dominant negative version of Gαi1 in the previous study.

Comparison of the structure of Apo-CXCR3 with CXCL10-CXCR3 structure reported recently[57] (PDB: 8K2X), reveals an overall RMSD of ~1 Å across the main-chain Cα atoms (Fig. 6n). The overall similarity in

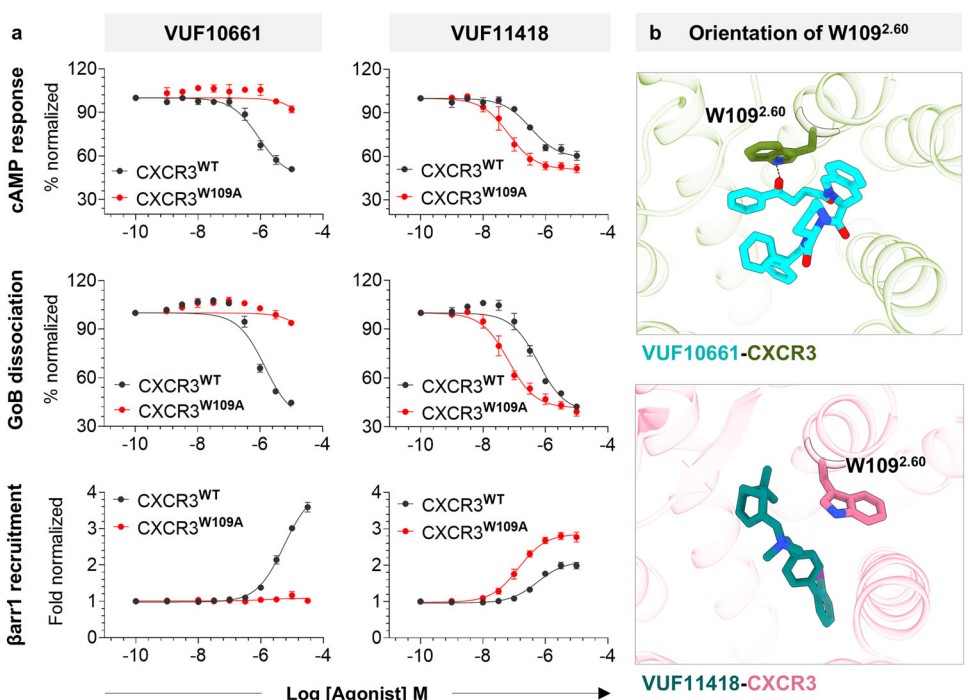

**Fig. 4 | W109$^{2.60}$ exerts a distinct effect in regulating signaling downstream to both VUFs. a** Stimulation of CXCR3$^{W109A}$ (red) with VUF10661 fails to elicit any response in G-protein signaling and βarr1 recruitment assays, while VUF11418 induces a slightly enhanced response, as compared to CXCR3$^{WT}$ (black). Data (mean ± SEM) represents three (for G-protein assays) or five (for βarr1 recruitment) independent experiments, performed in duplicate, and has been normalized with respect to the signal observed at the lowest ligand concentration, treated as either 100% (for G-protein assays) or 1 (for βarr1 recruitment). **b** W109$^{2.60}$ in CXCR3 interacts with VUF10661, while rotation of this residue in the VUF11418-bound structure prevents its interaction with VUF11418. (VUF10661-CXCR3: olive drab, VUF10661: cyan, VUF11418-CXCR3: pale violet red, VUF11418: deep teal). Source data are provided as a Source Data file.

the arrangement and conformation of the cytoplasmic side of the transmembrane helices (e.g., outward movement of TM5 and 6) in the two structures suggest that the Apo-CXCR3 structure reported here represents an active state of the receptor. However, we cannot conclude whether the active-like structure results from priming by CXCL10 used during purification and reconstitution of the complex, the allosteric stabilization of the active state by the bound G-protein, or a combination of both. Still, there are clear differences in the orthosteric binding pocket, in terms of the positioning of the extracellular side of the transmembrane helices and the conformation of CXCL10-interacting residues (Fig. 6o–t). In addition, the side chains of several CXCL10-interacting residues are not well resolved in the Apo-CXCR3 structure, which further corroborates our interpretation regarding the absence of CXCL10 in the orthosteric binding pocket.

### Agonist-induced activation and interaction of CXCR3 with G-proteins

Structural superimposition of the VUF10661/VUF11418-bound structures with the inactive state antagonist-bound structure of CXCR3 determined recently[57] (AMG487 bound CXCR3; PDB: 8K2W) (Fig. 7a) reveals the typical hallmarks of GPCR activation. Compared to the inactive conformation of CXCR3 our structures exhibit movements in the extracellular regions of TM1 and TM2, and intracellular regions of TM6 and TM7, by 3–5 Å, along with a linear shift of helix8 by approximately 3 Å toward the cytoplasmic core of the receptor (Fig. 7b, c). In addition, significant rotameric shifts are observed in the hydrophobic interaction network formed by the conserved triad residues Ala139$^{3.40}$-Pro227$^{5.50}$-Phe264$^{6.44}$, a variant of the Ile$^{3.40}$-Pro$^{5.50}$-Phe$^{6.44}$ motif, that stabilizes the inactive conformation of CXCR3 in the antagonist-bound state by restricting the mobility of TM5 and TM6 (Fig. 7d). Moreover, a series of conformational rearrangements are also observed in the other microswitches such as the DRY, NPxxY, and

CWxP motifs that are hallmarks of GPCR activation (Fig. 7d). The interaction interface of CXCR3 with G-protein harbors an approximate buried surface area of 3900 Å$^2$, and as anticipated, the distal end of the α5 helix inserts into the cytoplasmic side of the receptor transmembrane core (Supplementary Fig. 7d). The interface on the receptor side involves the cytoplasmic ends of TM2, TM3, TM5, and TM6, ICL2, ICL3 and helix8, while on the G-protein side, it involves the α1–β1 loop, β2–β3 loop, α2–β4 loop, and α5 helix of miniGαo (Supplementary Data 1). The carboxyl-terminus of α5 helix in miniGαo makes extensive contacts with the hydrophobic surface formed by the cytoplasmic ends of TM3, TM5, and TM6 of CXCR3 including hydrogen bonding between Ala345, Asn347, and Cys351 of α5 with Arg249$^{6.29}$, Asn152$^{3.53}$, and Arg149$^{3.50}$ of CXCR3, respectively (Supplementary Data 1). The terminal hook residue Tyr354 of α5 helix engages in hydrogen bonding with Val323 of helix8 in CXCR3. While ICL1 does not directly interact with G-proteins, ICL2 interacts with α5 helix, β2–β3 loop, and αN of miniGαo including a hydrogen bond formation between Ala31 of αN helix in miniGαo with Arg162 of ICL2 in CXCR3. Additionally, a number of ionic and hydrophobic interactions also help to further stabilize the receptor-G-protein interface that is listed in Supplementary Data 1.

### Allosteric network in CXCR3 driving transducer-coupling bias

In order to investigate the basis of ligand binding induced conformational rearrangements in CXCR3 which result in signaling bias, we compared the VUF11418 and VUF10661-bound structures (Fig. 8a). Interestingly, superimposition of the VUF-bound CXCR3 structures with each other reveals an RMSD of -1.2 Å. On binding the bulky VUF10661, Trp109$^{2.60}$ undergoes a rotameric transition of 180° away from the ligand binding pocket to allow the optimal positioning of the ligand. This rotameric shift makes space for the inward movement of the upper portion of TM1, which in turn facilitates the translation of Tyr60$^{1.39}$ by -3 Å toward the receptor core. This movement is then

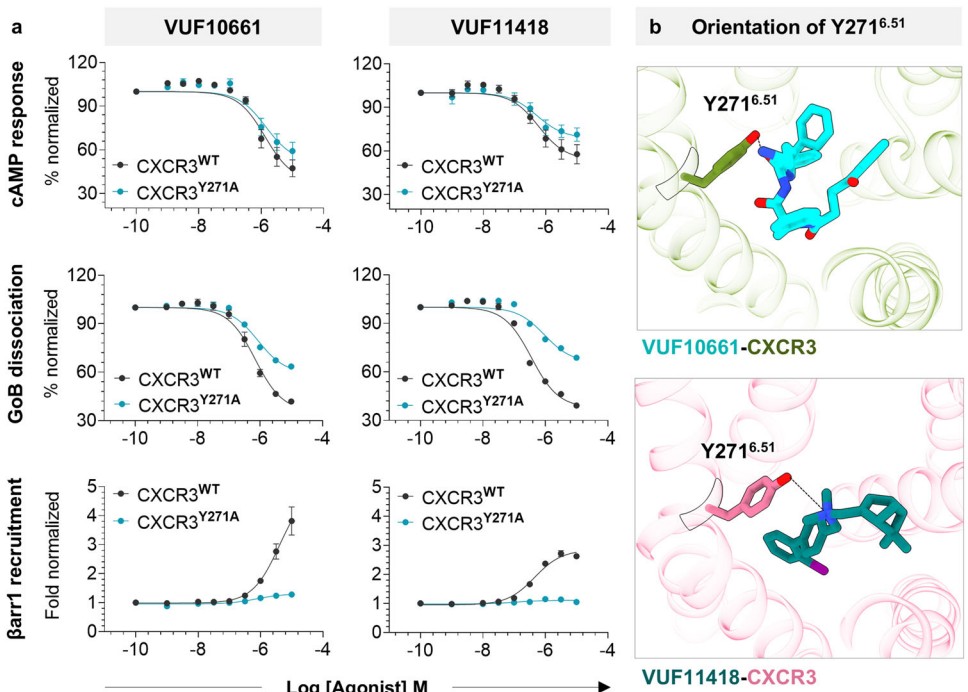

**Fig. 5 | Mutating Y271[6.51] leads to differential loss of signaling. a** Stimulation of CXCR3[Y271A] (blue) with either VUF10661 or VUF11418 induces G-protein signaling to a similar extent as wild-type receptor (black) when measured using a cAMP response assay, and reduced G-protein dissociation. Stimulation of CXCR3[Y271A] with VUF10661 and VUF11418 leads to a drastic loss in βarr1 recruitment. Data (mean ± SEM) represents either three (for G-protein assays) or four (for βarr1 recruitment) independent experiments, performed in duplicate, and has been normalized with respect to the signal observed at the lowest ligand concentration, treated as either 100% (for G-protein assays) or 1 (for βarr1 recruitment). **b** Y271[6.51] adopts a similar orientation and contacts the ligand in both VUF11418 and VUF10661-bound CXCR3 structures. (VUF10661-CXCR3: olive drab, VUF10661: cyan, VUF11418-CXCR3: pale violet red, VUF11418: deep teal). Source data are provided as a Source Data file.

relayed onto Tyr308[7.43], which in turn undergoes an angular shift of ~60° allowing for an inward movement of TM7 by ~1.8 Å. This permits a rotation of Trp268[6.48] by 80° toward the ligand binding pocket in the case of VUF10661 unlike in VUF11418 (Fig. 8b, c). These stark differences in CXCR3 upon binding of VUF10661 vs. VUF11418 hint at an allosteric network connecting the extracellular side of the receptors to the intracellular side through the transmembrane region that directs signaling bias exhibited by these agonists. However, we would like to note that a precise delineation of the allosteric network driving signaling bias will require additional structural snapshots, particularly in complexes with βarrs.

### Dual-agonism of CXCR3 agonists at CXCR7

The only other C-X-C type chemokine receptor for which small molecule agonists have been described is CXCR7[47], an exclusively βarr-biased receptor, with no measurable G-protein signaling[17,18] (Fig. 9a). Taking this into consideration, we compared our structures with the recently published structure of CXCR7 in complex with a small molecule partial agonist CCX662[58]. Superimposition of VUF10661-CXCR3 and CCX662-CXCR7 reveals an overall RMSD of ~1 Å along the main-chain Cα atoms of the receptor, and in contrast to a "U-shaped" conformation of VUF10661, CCX662 adopts a "curved and shallow" conformation on CXCR7 (Supplementary Fig. 8a). Both the ligands exhibit an overall similar positioning and corresponding interacting residues in the two receptors. However, they differ in terms of the relatively deeper binding of CCX662 in CXCR7 and exhibit differential orientation of some of the key interacting residues (Supplementary Fig. 8b–e). Additionally, differential positioning of ICL1, 2, and helix8 is evident between the two structures (Supplementary Fig. 8f). It should be noted that the differences observed on the cytoplasmic side of the receptors might originate from the interaction of the receptors with G-protein (CXCR3) vs. a stabilizing antibody fragment (CXCR7). Similarly, the

structure of VUF11418-CXCR3 also superimposes well with the CCX662-CXCR7 structure, with an RMSD of ~1 Å along the main-chain Cα atoms of the receptor, with both the ligands occupying an overall similar pocket (Supplementary Fig. 8g). However, there is a significant lateral shift between the two ligands in the binding pocket relative to each other, which leads to differential interaction with the residues in the corresponding receptors (Supplementary Fig. 8h, i).

Interestingly, upon comparing the key residues in the orthosteric binding pocket of the two receptors, we observed significant conservation of these residues between the two receptors (Fig. 9b, c), and this prompted us to probe the reactivity of VUF11418 and VUF10661 on CXCR7, and by extension, on the entire panel of CXCRs (Fig. 9d). Surprisingly, we observed that both VUFs are robust agonists for CXCR7 in βarr recruitment while being silent on G-protein-coupling assays (Fig. 9d). Using a previously characterized small molecule agonist of CXCR7, namely VUF11207 as a reference[18], we further confirmed that VUF11418 and VUF10661 are strong agonists at CXCR7 (Fig. 10a and Supplementary Fig. 8j). These data suggest that in contrast to previously believed notion, small molecule agonists VUF11418 and VUF10661 are dual agonists of CXCR3 and CXCR7. It is interesting to note that both CXCR3 and CXCR7 share a common natural chemokine agonist CXCL11 as well, which has been previously extensively characterized[36,38]. Our findings with VUF11418 and VUF10661 demonstrate that ligand promiscuity encoded in the natural chemokine agonists can also be recapitulated by engaging only the orthosteric binding pocket by small molecules. At the same time, the exclusive selectivity of VUF11207 for CXCR7 underscores that selective targeting of the chemokine receptors is also possible, and our structural templates provided here may facilitate efforts in this direction.

Interestingly, mutation of the ligand interacting residues on CXCR7, that are conserved with CXCR3, to alanine also exhibited varying degrees of loss in βarr recruitment, upon stimulation with

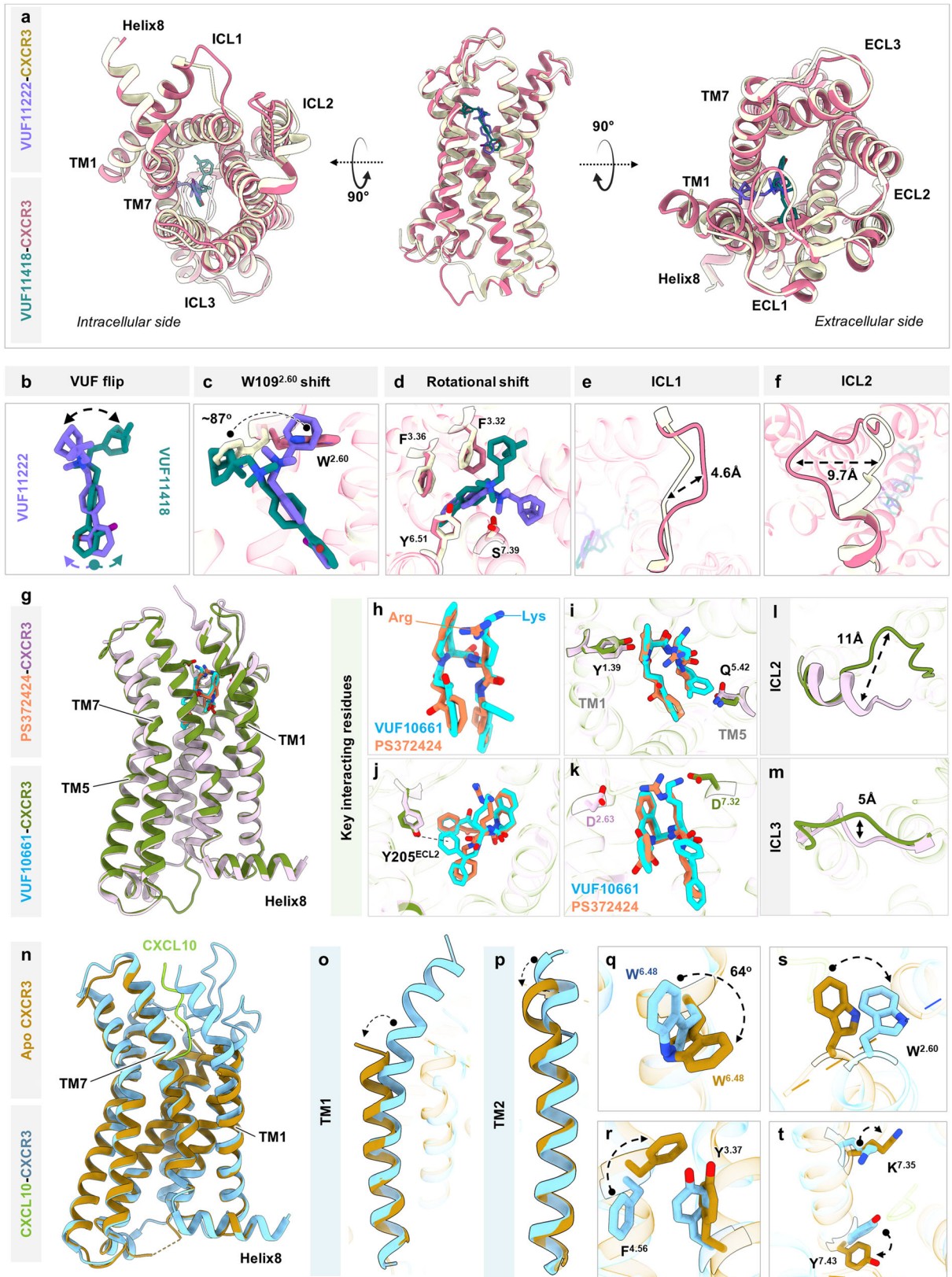

either VUF10661 or VUF11418 (Fig. 10b). Intriguingly, one of the mutants, CXCR7$^{L128A}$ exhibited differential response following stimulation with the two VUFs wherein stimulation with VUF10661 lead to a near-complete loss of βarr recruitment, while VUF11418 is still able to induce robust βarr recruitment, albeit at a slightly lower level as compared to the wild-type receptor (Fig. 10c). On the other hand,

stimulation of CXCR7$^{Y268A}$ with both the ligands lead to a dramatic loss of βarr recruitment, which is analogous to that observed for the corresponding mutation in CXCR3 (i.e., Y271$^{6.51A}$) (Fig. 10c). A direct structural comparison of CXCR3 and CXCR7 structures suggests that the local conformation of the key residues in CXCR3 engaged in interaction with VUF10661, the βarr-biased agonist, align well with the

**Fig. 6 | Comparative analysis of CXCR3 structures. a** Bottom view (left), side view (middle), and top view (right) of CXCR3 depicting structural reorientation upon binding VUF11418 (deep teal) and VUF11222 (violet). **b** Flip observed in VUF11418 and VUF11222 to contact nearby CXCR3 residues. **c** Rotational shift in W109[2.60] to avoid a steric clash with the small molecule agonists. **d** Rotational shift in the side chains of the residues constituting the aromatic cage around the ligand. **e** Shift of ICL1 away from the cytoplasmic core in VUF11418-bound CXCR3. **f** Opening up of helical conformation in ICL2 of VUF11418-bound CXCR3 and movement toward the cytoplasmic core. **g** Structural superimposition of VUF10661-CXCR3 and PS372424-

CXCR3. **h–k** Major interacting residues in CXCR3 contacting VUF10661 (cyan) and PS372424 (coral). **l, m** Shift in ICL2 (**l**) and ICL3 (**m**) of VUF10661-bound CXCR3 as compared to PS372424-bound CXCR3. **n** Structural superimposition of Apo-CXCR3 and CXCL10-bound CXCR3. **o, p** Transmembrane helices move away from the orthosteric pocket in the Apo-CXCR3 structure. **q–t** Side chain rearrangements in the key residues as observed in Apo-CXCR3 and CXCL10-CXCR3. (VUF11418-CXCR3: pale violet red, VUF11222-CXCR3: light beige; PDB: 8HNM, VUF10661-CXCR3: olive drab, PS372424-CXCR3: pale violet; PDB: 8HNL, Apo-CXCR3: dark goldenrod, CXCL10-CXCR3: sky blue; PDB: 8K2X, CXCL10: chartreuse).

corresponding residues in CXCR7 (Fig. 10d). Considering the intrinsic βarr-bias of CXCR7, these observations support the dual-agonism exhibited by VUF10661 and VUF11418 as well as the contribution of allosteric network and associated local conformational changes in directing transducer-coupling bias at these receptors.

## Discussion

We note that a pair of recent studies have reported the cryo-EM structures of CXCR3 in a complex with G-proteins and another small molecule agonist[56,57]. The overall structural features observed in these contemporary studies align well with each other although there are notable differences in ligand positioning despite similar chemical structures. While it is possible that small molecule agonists, even with minor structural differences, can be accommodated in different poses in the orthosteric binding pocket, it will be interesting to explore in future studies if these differences correspond to the distinct efficacy of the ligands. The dual-agonism observed for VUF10661 and VUF11418 on CXCR3 and CXCR7 is intriguing, especially considering that they do not activate any of the other five CXCRs in our functional assays. It is tempting to speculate that the structural features of the orthosteric binding pocket between CXCR3 and CXCR7 allow the dual-agonism of these agonists, reminiscent of their ability to recognize a shared chemokine agonist, CXCL11. However, it would be interesting to explore in future studies if these agonists can be tailored further to expand their cross-reactivity to other CXCRs or make them CXCR3 specific, and it may provide further insights into ligand promiscuity at these chemokine receptors.

It is also worth noting that based on radioligand binding experiments, previous studies have reported VUF10661 to bind to an allosteric site on CXCR3[46], based on the observation that this small molecule enhanced the maximal binding of radiolabeled chemokines but did not display competitive binding. Considering similar findings in the context of other chemokine receptor allosteric ligands, it was proposed to occupy an allosteric site. Although our structures reveal the binding of VUF ligands to the orthosteric site of the receptor, typically occupied by the chemokines, it is also plausible that VUF ligands and chemokines can bind simultaneously. Future studies are essential to probe the interesting pharmacology of the VUF ligands, especially their modulation of chemokine binding to CXCR3, using biochemical and structural approaches, to better understand their interaction with the receptor.

Taken together, our findings provide molecular insights into small molecule recognition and activation of CXCR3 as well as uncover dual-agonism at CXCR3 and CXCR7. These findings pave the way for a better understanding of agonist recognition and biased signaling encoded in the chemokine receptors from a therapeutic perspective.

## Methods

### General plasmids, reagents, and cell culture

Most of the molecular biology and general reagents were purchased from Sigma-Aldrich unless mentioned otherwise. Dulbecco's Modified Eagle's Medium (DMEM), Phosphate buffered saline (PBS), Fetal-Bovine Serum (FBS), Hank's balanced salt solution (HBSS), Trypsin-EDTA and penicillin-streptomycin solution were purchased from

Thermo Fisher Scientific. HEK293 cells (purchased from ATCC, Cat. no: CRL-3216) were maintained in 10 cm dishes (Corning, Cat. no: 430167) at 37 °C under 5% $CO_2$ in Dulbecco's Modified Eagle's Medium (Gibco, Cat. no: 12800-017) supplemented with 10% FBS (Gibco, Cat. no: 10270-106), 100 U/mL penicillin and 100 μg/mL streptomycin (Gibco, Cat. no: 15140-122). *Sf9* cells (purchased from Expression Systems, Cat. no: 94-001 F) were maintained in either ESF921 media (Expression Systems, Cat. no: 96-001-01) or Sf-900™ II SFM serum-free media (Gibco, Cat. no: 10902088). Lauryl Maltose Neopentyl Glycol (L-MNG) was purchased from Anatrace (Cat. no: NG310). The coding regions for CXCR1-7 were sub-cloned in both pcDNA3.1 vector (with an N-terminal FLAG-tag) as well as pCAGGS vector (with an N-terminal FLAG-tag and a C-terminal SmBiT fusion). CXCR3 was also sub-cloned in the pVL1393 vector (with an N-terminal FLAG-tag) followed by the N-terminal region of the M4 receptor (residues 2–23) which was then used to generate baculovirus encoding CXCR3. The constructs used for NanoBiT-based assays were previously described[59]. All DNA constructs were confirmed by sequencing from Macrogen. CXCL10 was purified as previously described[60], while VUF11418 and VUF10661 were synthesized and characterized as previously described[3,48]. The sequences for the various primers used in this study have been provided in Supplementary Table 1.

### Purification of CXCR3

Full-length recombinant CXCR3 was isolated from *Spodoptera frugiperda (Sf9)* insect cells following a previously published protocol[51–53]. *Sf9* cells were harvested 72 h post-infection with CXCR3 expressing baculovirus. This was followed by homogenization of the cells initially in hypotonic buffer (20 mM HEPES pH 7.4, 20 mM KCl, 10 mM $MgCl_2$, 1 mM PMSF, 2 mM benzamidine) and subsequently in hypertonic buffer (20 mM HEPES pH 7.4, 20 mM KCl, 10 mM $MgCl_2$, 1 M NaCl, 1 mM PMSF, 2 mM benzamidine). Cells were then subjected to solubilization by incubating in lysis buffer (20 mM HEPES pH 7.4, 450 mM NaCl, 1 mM PMSF, 2 mM benzamidine, 0.1% cholesteryl hemisuccinate, 2 mM iodoacetamide) and 0.5% L-MNG (Anatrace, Cat. no: NG310) for 2 h at 4 °C. Next, the lysate was diluted in 2× volume of dilution buffer (20 mM HEPES pH 7.4, 2 mM $CaCl_2$, 1 mM PMSF, and 2 mM benzamidine) to reduce the salt concentration to 150 mM NaCl. Debris were removed by centrifuging the lysate at $47,936 \times g$ for 30 min. The supernatant was filtered and loaded onto pre-equilibrated M1-FLAG beads. The column was then washed alternatively with LSB/ Low Salt Buffer (20 mM HEPES pH 7.4, 150 mM NaCl, 2 mM $CaCl_2$, 0.01% cholesteryl hemisuccinate, 0.01% L-MNG) and HSB/ High Salt Buffer (20 mM HEPES pH 7.4, 350 mM NaCl, 2 mM $CaCl_2$, 0.01% L-MNG). Protein was eluted in the presence of 2 mM EDTA and 250 μg/mL FLAG. To prevent receptor aggregation, free cysteines were blocked by incubating with 2 mM iodoacetamide. Excess-free iodoacetamide was quenched by incubating with 2 mM L-cysteine. Ligand (either 100 nM CXCL10 or 1 μM VUF11418 or 1 μM VUF10661 was kept in all the buffers). Ligand-bound receptors were stored in the presence of 10% glycerol at −80 °C till further use.

### Purification of G-proteins

MiniGαo was purified from *E. coli* BL21 (DE3) cells according to a previously published protocol[51,52]. The construct used here contains an

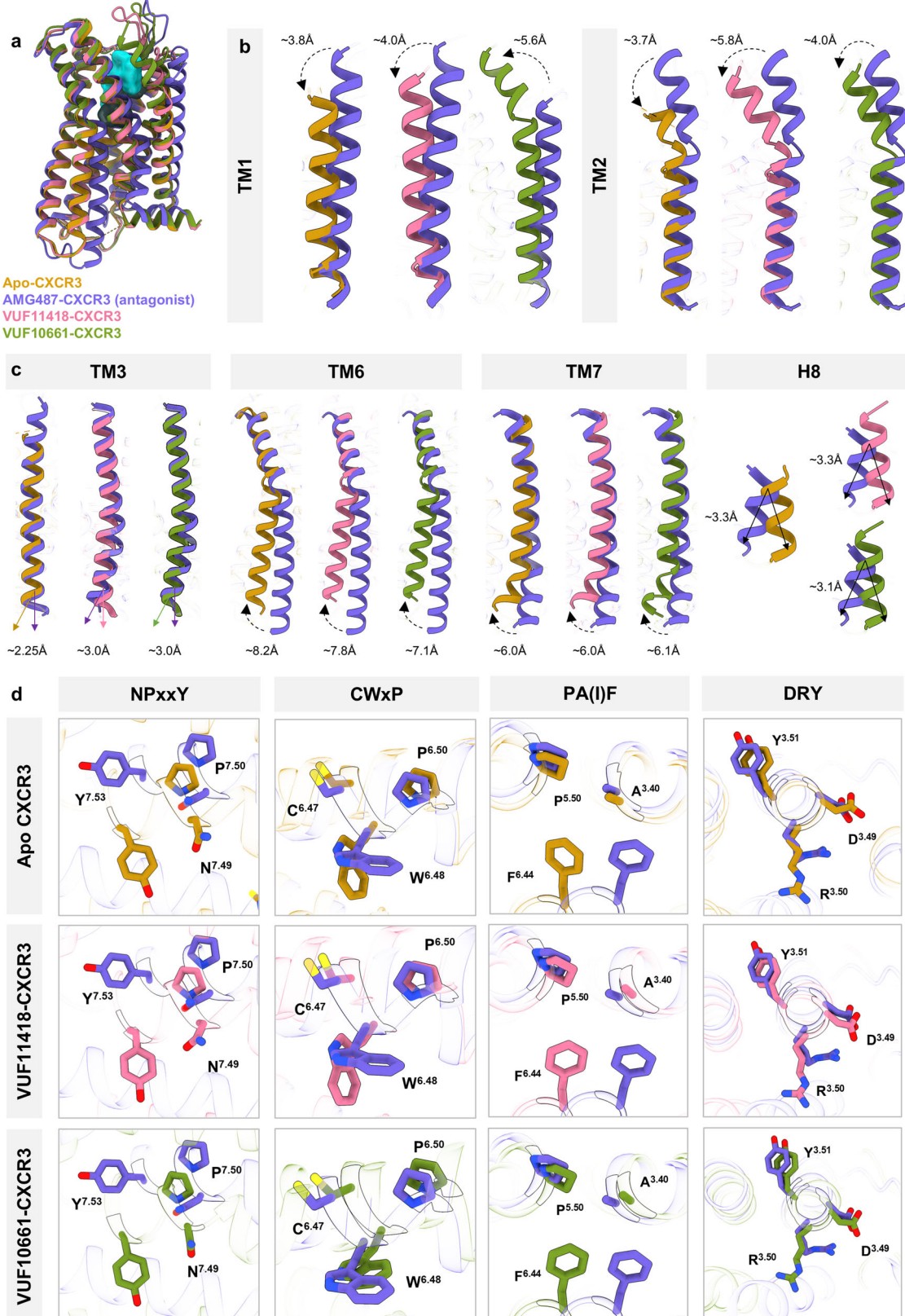

**Fig. 7 | Major conformational changes on CXCR3 activation. a** Superimposition of inactive CXCR3 with Apo-state receptor and receptor bound to VUF11418 and VUF10661. **b**, **c** Displacements of TM1, TM2, TM3, TM6, TM7 and helix8 upon CXCR3 activation in the structures of Apo-CXCR3, VUF11418-CXCR3, VUF10661-CXCR3, respectively. **d** Conformational changes in the conserved microswitches (DRY, PA(I)F, NPxxY, CWxP) in the structures of CXCR3. (Apo-CXCR3: dark goldenrod, VUF11418-CXCR3: pale violet red, VUF10661-CXCR3: olive drab, AMG487-CXCR3: purple; PDB: 8K2W).

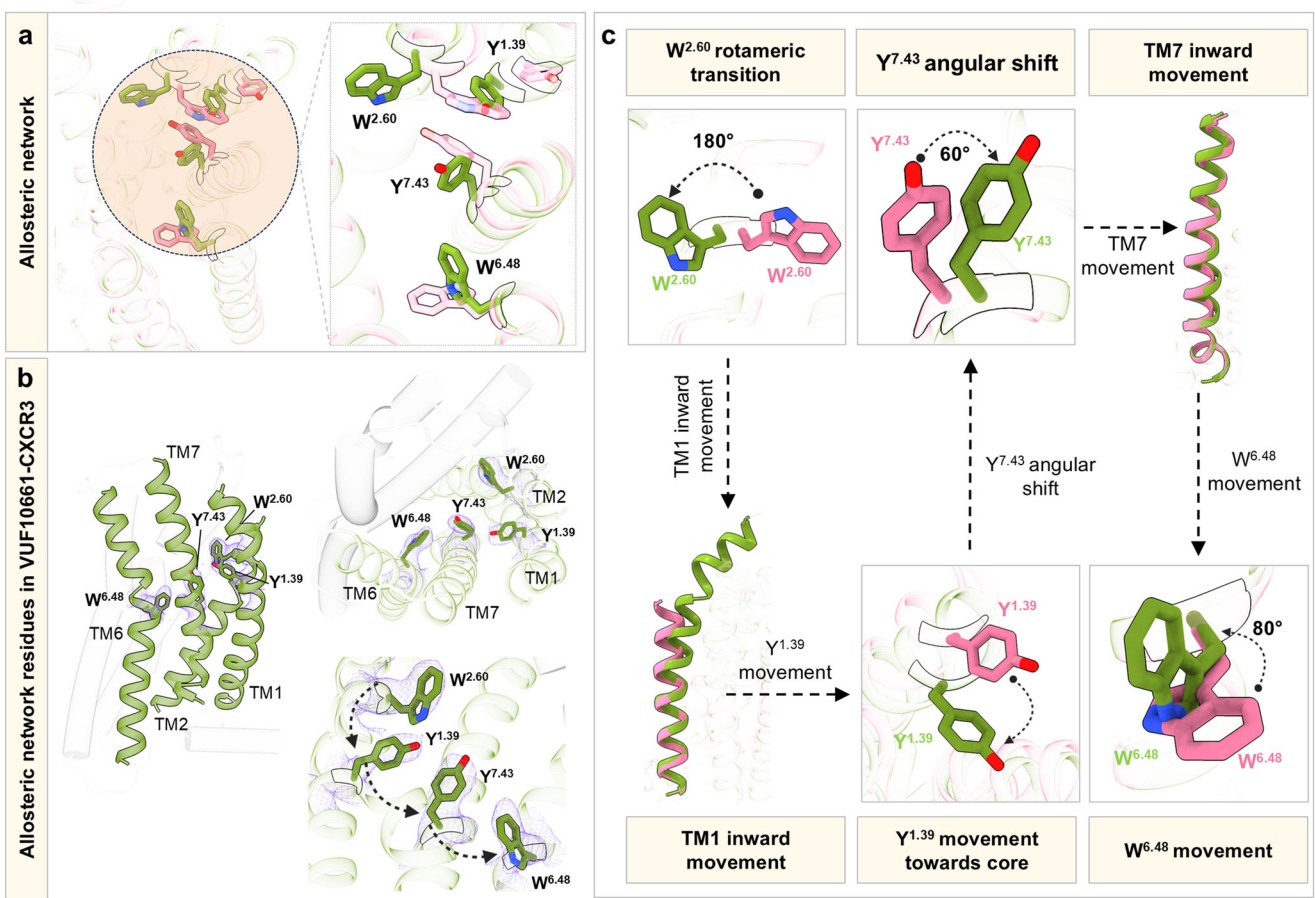

**Fig. 8 | Allosteric network in VUF10661-bound CXCR3. a** Residues promoting allosteric communication in VUF10661-CXCR3 exhibit different orientations than those in VUF11418-CXCR3. Superimposition of the two VUF-bound structures reveals an RMSD of ~1.2 Å. **b** Key residues in CXCR3 promoting allosteric communication. **c** Allosteric signal propagation in VUF10661-CXCR3. (VUF11418-CXCR3: pale violet red, VUF10661-CXCR3: olive drab).

N-terminal hexa-histidine tag followed by a TEV protease cleavage site, and a GGSGGSGG linker sequence replacing the His[57]-Thr[182] of the α-helical domain. A starter culture was grown for 6–8 h at 37 °C in LB media, followed by an overnight primary culture at 30 °C in the presence of 0.2% glucose supplementation. The secondary culture was grown in TB/ Terrific Broth media and induced at an $OD_{600}$ of 0.8 with 50 μM IPTG. Following induction, cells were cultured for an additional 18–20 h at 25 °C. Cells thus obtained were lysed by sonication in lysis buffer (40 mM HEPES pH 7.4, 100 mM NaCl, 10% Glycerol, 10 mM Imidazole, 5 mM $MgCl_2$, 1 mM PMSF, 2 mM benzamidine, 1 mg/mL lysozyme, 50 μM GDP and 100 μM DTT). Cell debris was removed by centrifuging at $47,936 \times g$ for 30 min and the filtered supernatant was enriched on Ni-NTA beads. Excess unbound protein was removed by washing with wash buffer (20 mM HEPES pH 7.4, 500 mM NaCl, 40 mM Imidazole, 10% Glycerol, 50 μM GDP, and 1 mM $MgCl_2$) and bound protein was eluted in 20 mM HEPES pH 7.4, 100 mM NaCl, 10% Glycerol and 500 mM Imidazole. 6X-His-tag was removed by treating with TEV protease overnight (ratio of TEV:protein was kept at 1:20) at room temperature and cleaved untagged protein was isolated by size exclusion chromatography using HiLoad Superdex 200 PG 16/600 column (Cytiva, Cat. no: 28989335). Fractions corresponding to our protein of interest were pooled, quantified, and stored in the presence of 10% glycerol at −80 °C till further use.

Gβ1γ2 was purified from *Sf9* insect cells as previously described[51,52]. Gβ1 and Gγ2 were co-expressed in *Sf9* insect cells using the baculovirus expression system, with Gβ1 containing an N-terminal His tag. 72 h post-infection, cells were harvested and lysed by sequentially douncing first in lysis buffer (20 mM Tris-Cl pH 8.0, 300 mM NaCl, 10% Glycerol, 1 mM PMSF, 2 mM benzamidine and 1 mM $MgCl_2$) and then in solubilization buffer (20 mM Tris-Cl pH 8.0, 300 mM NaCl, 10% Glycerol, 1% DDM, 5 mM β-ME, 10 mM Imidazole, 1 mM PMSF and 2 mM benzamidine). Solubilization was allowed to proceed for 2 h at 4 °C, which was followed by centrifugation at $47,936 \times g$ for 30 min to clear cellular debris. The supernatant was filtered and loaded onto pre-equilibrated Ni-NTA beads. Unbound protein was removed by washing extensively with wash buffer (20 mM Tris-Cl pH 8.0, 300 mM NaCl, 30 mM Imidazole, 10% glycerol, 5 mM β-ME and 0.02% DDM (Anatrace, Cat. no: D310A)) and eluted with 20 mM Tris-Cl pH 8.0, 300 mM Imidazole and 0.01% L-MNG. Eluted protein was quantified and stored in the presence of 10% glycerol at −80 °C till further use.

**Purification of scFv16**
Gene encoding scFv16 was cloned in pET-42a (+) vector with an in-frame N-terminal 10X-His-MBP tag followed by a TEV cleavage site and expressed in *E. coli* Rosetta (DE3) strain, following a previously published protocol[51,52]. Overnight primary culture was transferred to 1 L 2XYT media supplemented with 0.5% glucose and 5 mM $MgSO_4$. The culture was then induced at an $OD_{600}$ of 0.9 with 250 μM isopropyl-β-D thiogalactopyranoside (IPTG) and allowed to grow for 16–18 h at 18 °C. Cells were harvested and resuspended in 20 mM HEPES pH 7.4, 200 mM NaCl, 10 mM Imidazole, 2 mM Benzamidine, and 1 mM PMSF and incubated at 4 °C for 40 min with constant stirring. Cells were disrupted by ultrasonication and cell debris was removed by centrifugation at $38,828 \times g$ for 40 min at 4 °C. Protein was enriched on Ni-NTA resins, and beads were washed extensively with 20 mM HEPES pH

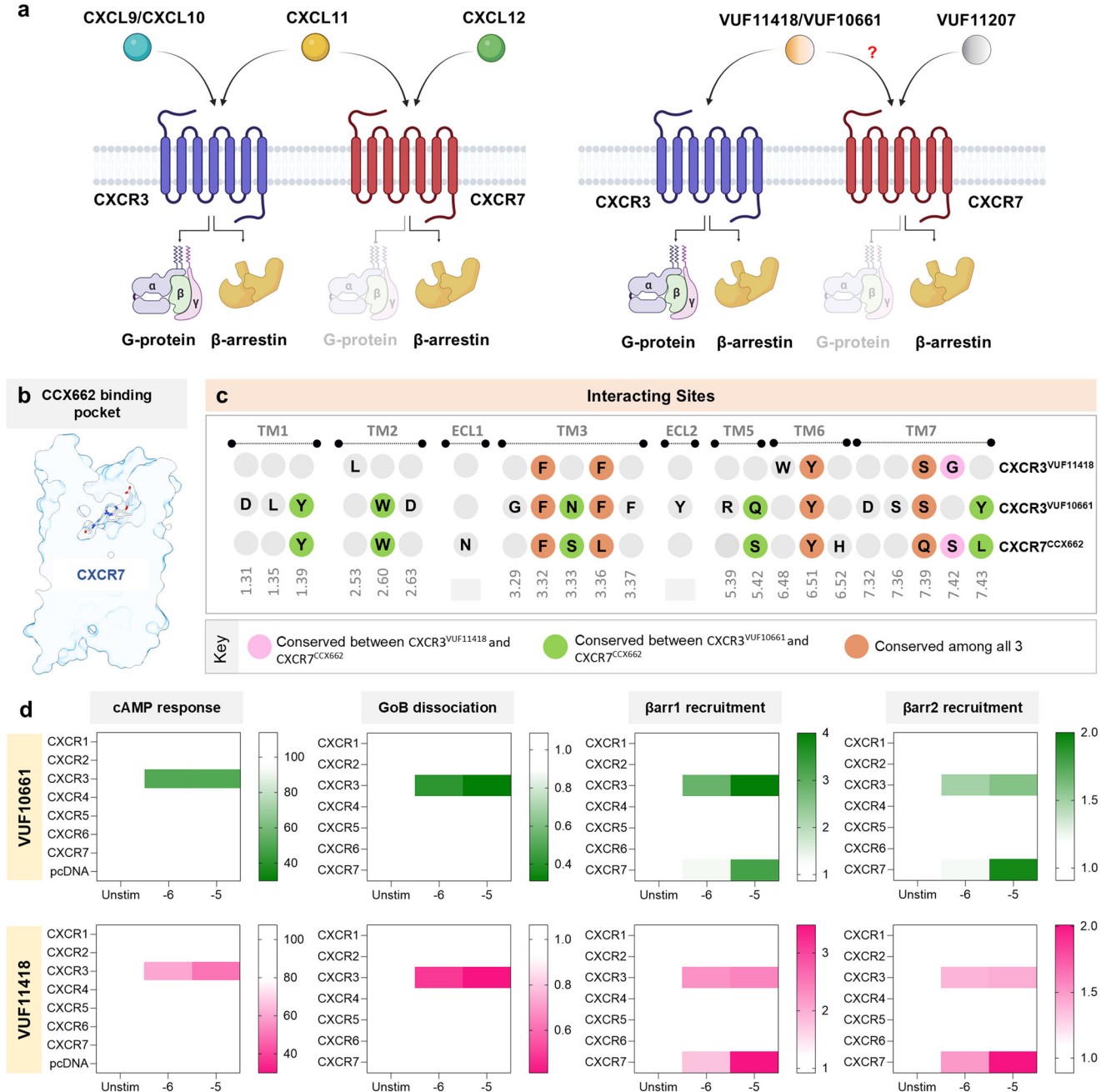

**Fig. 9 | VUF11418 and VUF10661 activate signaling downstream to CXCR7.**
**a** Schematic representation of canonical signaling downstream of CXCR3 and
CXCR7 stimulation with chemokines and synthetic agonists. Created in BioRender.
#3, G. (2025) https://BioRender.com/r69j488. **b** Cross-section of the ligand binding
pocket in CCX662 bound CXCR7 (PDB: 7SK9). **c** Conserved interacting sites in
VUF11418-CXCR3, VUF10661-CXCR3 and CCX662-CXCR7. **d** Heatmap showing
VUF10661 (green) and VUF11418 (pink) selectivity across all CXCRs in inducing

cAMP signaling (Gi-mediated decrease in cytosolic cAMP), GoB dissociation (as
measured by a decrease in luminescence) and βarr1/2 recruitment (as measured by
an increase in luminescence). Data (mean) represents three independent biological
replicates, performed in duplicate, and normalized with respect to signal observed
in the absence of stimulation, treated either as 100% (for cAMP response), or 1 (for
GoB dissociation and βarr1/2 recruitment). Source data are provided as a Source
Data file.

7.4, 200 mM NaCl, 10 mM Imidazole. Bound protein was eluted with
300 mM Imidazole in 20 mM HEPES pH 7.4, 200 mM NaCl. Subse-
quently, the Ni-NTA elute was enriched on amylose resin (NEB, Cat. no:
E8021L) and washed with 20 mM HEPES pH 7.4, and 200 mM NaCl to
remove non-specific proteins. Protein was then eluted with 10 mM
maltose prepared in 20 mM HEPES pH 7.4, 200 mM NaCl, and the His-
MBP tag was removed by overnight treatment with TEV protease (ratio
of TEV:Protein was kept at 1:20). Tag-free scFv16 was recovered by
passing TEV-cleaved protein through Ni-NTA resin. Eluted protein was
concentrated with Vivaspin 10 kDa MWCO concentrator (Cytiva Life

sciences, Cat. no: 28932360) and cleaned by size exclusion chroma-
tography on HiLoad Superdex 16/600 200 PG column (Cytiva Life
sciences, Cat. no: 28989335). Fractions corresponding to scFv16 were
pooled, flash-frozen, and stored at −80 °C in the presence of 10%
glycerol.

### Reconstituting chemokine/synthetic ligand-chemokine receptor-G-protein complexes

Purified ligand-receptor complex was incubated with a 1.2-fold molar
excess of miniGαo, Gβ1γ2, and scFv16, in the presence of 5 mM CaCl₂

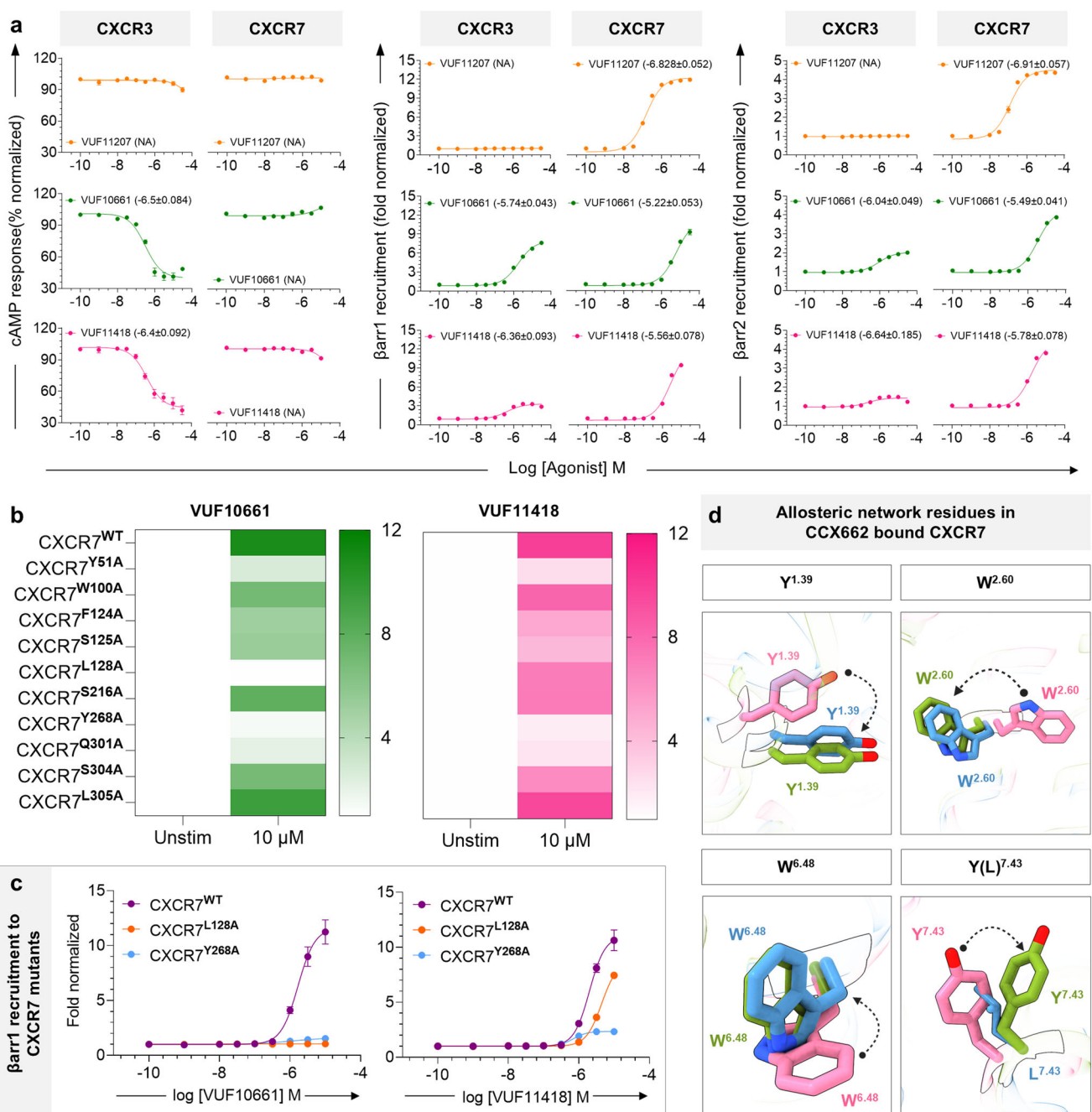

**Fig. 10 | Validation of dual-agonism of VUFs at CXCR7. a** VUF10661 (green) and VUF11418 (pink) stimulate both CXCR3 and CXCR7 as measured in various assays. Data (mean ± SEM) represents either three (for βarr1/2 recruitment) or four (for cAMP response assay) independent biological replicates, performed in duplicate, and normalized with respect to signal observed at the lowest dose, treated either as 100% (for cAMP response), or 1 (βarr1/2 recruitment). VUF11207 (orange) has been previously characterized to be specific for CXCR7. **b** Heatmap showing βarr1 recruitment downstream to CXCR7 mutants following stimulation with VUF10661 (green) and VUF11418 (pink). Data (mean ± SEM) represents three independent experiments, performed in duplicate, and has been normalized with respect to the signal observed under unstimulated condition, treated as 1. **c** Mutating Y268A

(blue) leads to a drastic reduction in βarr1 recruitment following stimulation with both VUFs, whereas mutating L128A (orange) ablates βarr1 recruitment upon stimulation with VUF10661 while eliciting only marginally reduced βarr1 recruitment following VUF11418 stimulation, as compared to wild-type CXCR7 (purple). Data (mean ± SEM) represents four independent experiments, performed in duplicate, and has been normalized with respect to the signal observed under unstimulated conditions, treated as 1. **d** Residues promoting allosteric communication in VUF10661-CXCR3 (green) exhibit similar rotameric shifts with respect to CCX662-CXCR7 (blue, PDB: 7SK9) and different orientations than those in VUF11418-CXCR3 (pink). Source data are provided as a Source Data file.

and 25mU/mL apyrase (NEB, Cat. no: M0398S), for 2 h at room temperature. The mixture was then concentrated using a 100 MWCO concentrator (Cytiva, Cat. no: GE28-9323-19) and injected into the Superdex 200 Increase 10/300 GL SEC column to separate the receptor-G-protein complex from the free components. Peak fractions were analyzed by running an SDS-PAGE. Fractions containing the complex were pooled and concentrated to roughly 12-18 mg/mL using the same concentrator and stored at −80 °C until further use. SEC profile and SDS-PAGE of the various complexes is provided in Supplementary Fig. 1.

## Negative stain electron microscopy

Prior to grid freezing for high-resolution cryo-EM data collection, conventional uranyl formate negative staining was used to assess sample homogeneity[53,61]. In brief, 3.5 μL of the sample was dispensed on a formvar/carbon-coated 300 mesh copper grid (PELCO, Ted Pella), incubated for 1 min, and then blotted off using Whatman No. 1 filter paper. The grid containing the attached sample was then touched onto a drop of freshly prepared 0.75% uranyl formate stain, which was immediately blotted off using filter paper. To improve staining efficiency, the grid was then placed on a second drop of uranyl formate and moved in a rotating fashion for 30 s. Before imaging and data collection, the excess stain was blotted off and allowed to air dry. A FEI Tecnai G2 12 Twin TEM (LaB6) operating at 120 kV and outfitted with a Gatan 4k × 4k CCD camera at 30,000× magnification was used for imaging and data collection. For further analysis, the acquired micrographs were imported into Relion 3.1.2[62–64]. About 10,000 particles were automatically selected using the Gaussian blob picker, extracted with a box size of 280 pix, and then submitted to reference-free 2D classification to obtain the final 2D class averages.

## Cryo-EM grid preparation and data collection

3.0 μL of the purified CXCR3-Go complexes were dispensed onto glow-discharged Quantifoil holey carbon grids (R1.2/1.3, Au, 300 mesh) at a concentration of approximately 13.2 mg/mL (VUF10661-CXCR3-Go), 18.5 mg/mL (VUF11418-CXCR3-Go) and 14.7 mg/mL (Apo-CXCR3-Go). The grids were blotted for 4 s at 4 °C and 100% humidity with a blot force of 10 using a Vitrobot Mark IV (Thermo Fischer Scientific) and immediately plunge frozen in liquid ethane (−181 °C).

Data collection of all samples was performed on a Titan Krios G3i (Thermo Fisher Scientific) operating at an accelerating voltage of 300 kV equipped with a Gatan K3 direct electron detector and Bio-Quantum K3 imaging filter. Movie stacks were acquired in counting mode at a pixel size of 0.83 Å/pix and a dosage rate of approximately 15.6e⁻/Å²/s using EPU software over a defocus range of -0.8 to −1.6 μm. Each movie was fractionated into 48 frames with a total dose of 50.1e⁻/Å² that was obtained throughout the 2.3 s exposure period. In total, 3165, 3030, and 3125 movie stacks were collected for VUF10661-CXCR3, VUF11418-CXCR3, and Apo-CXCR3 samples, respectively.

## Cryo-EM data processing

Movie stacks were aligned (4×4 patches) and dose-weighted using RELION's implementation of the MotionCor2 algorithm[62]. The motion-corrected micrographs were imported into cryoSPARC v4.4[65], and CTF parameters were estimated with Patch CTF (multi).

For the VUF10661-CXCR3-Go dataset, 1,384,864 autopicked particle projections were extracted using a box size of 280 pix (Fourier cropped to 70 pix) and then subjected to 2D classification for cleaning. 363,327 particle projections corresponding to 2D class averages with evident secondary features were selected, re-extracted with a box size of 280 pix (Fourier cropped to 180 pix), and subjected to heterogeneous refinement into 3 classes. The particles curated through several rounds of heterogeneous refinement were exported to RELION v4.0. Subsequently, further curation was performed, with a focus on the receptor region, followed by Bayesian polishing with a box size of 300 pix (Fourier cropped to 240 pix). The 116,462 resulting particles were imported into cryoSPARC and subjected to non-uniform refinement with estimating CTF parameters, yielding a reconstruction with a nominal resolution of ~3.0 Å at a Fourier shell correlation of 0.143. In order to improve the resolution and features corresponding to the receptor, local refinement was performed with a mask on the receptor, yielding a reconstruction with a nominal resolution of ~3.2 Å.

For the VUF11418-CXCR3-Go dataset, 1,527,953 particles were autopicked from 3030 motion-corrected micrographs using the template-picker subprogram within cryoSPARC. Picked particles were extracted with a box size of 280 pix and Fourier cropped to 70 pix, and subjected to 2D classification and heterogeneous refinement to remove ice contamination and dissociated particles. The resulting 360,223 particles were re-extracted with a box size of 280 pix (Fourier cropped to 180 pix) and subjected to heterogeneous refinement into 3 classes. The 182,526 resulting particles were exported to RELION v4.0. Subsequently, further curation was performed, with a focus on the receptor region, followed by Bayesian polishing with a box size of 300 pix (Fourier cropped to 240 pix). The 150,213 resulting particles were imported into cryoSPARC and subjected to non-uniform refinement with estimating CTF parameters, yielding a reconstruction with a nominal resolution of ~3.1 Å at a Fourier shell correlation of 0.143. Local refinement of the receptor region with a mask improved the density derived from the receptor, yielding a reconstruction with a nominal resolution of ~3.5 Å.

For the Apo-CXCR3-Go dataset, a template-picker was used to automatically pick particles from 3125 motion-corrected micrographs. The 1,633,141 picked particles were extracted with a box size of 280 pix (Fourier cropped to 70 pix) and subjected to 2D classification to remove the contaminations and dissociated particles. The resulting 298,771 particles were re-extracted with a box size of 280 pix and Fourier cropped to 180 pix followed by ab-initio reconstruction heterogeneous refinement. The 173,083 resulting particles were exported to RELION v4.0. Subsequently, further curation was performed, with a focus on the receptor region, followed by Bayesian polishing with a box size of 300 pix (Fourier cropped to 240 pix). The 41,722 resulting particles were imported into cryoSPARC and subjected to non-uniform refinement with estimating CTF parameters, yielding a reconstruction with a nominal resolution of 3.3 Å at a Fourier shell correlation of 0.143. To improve the resolution and features corresponding to the receptor, local refinement was performed with a mask on the receptor, yielding a reconstruction with a nominal resolution of ~3.7 Å.

Local resolution of all maps was calculated using Blocres included within the cryoSPARC package[65] with the half maps as input. Final maps were sharpened with phenix.auto_sharpen[66,67] to enhance features for model building.

## Model building and refinement

The initial model of CXCR3 was generated from the AlphaFold2 model (https://alphafold.ebi.ac.uk/entry/A0A0S2Z3W5), while the atomic coordinates of miniGαo, and other components of G-protein (Gβ1, Gγ2, and scFv16) were obtained from the cryo-EM structure of GALR1-miniGαo complex[68] (PDB: 7XJJ) and MT1-Gi complex[69] (PDB: 7DB6), respectively. Ligand coordinates and geometric restraints were generated with Grade web server (Smart, O.S., Sharff A., Holstein, J., Womack, T.O., Flensburg, C., Keller, P., Paciorek, W., Vonrhein, C. and Bricogne G. (2021) Grade2 version 1.5.0. Cambridge, United Kingdom: Global Phasing Ltd.). These initial models were roughly docked into the density maps using UCSF ChimeraX-1.5[70,71], followed by rigid body and flexible fitting of the coordinates with the jiggle fit and all-atom refine module in COOT 0.9.6[72]. DeepEMhancer maps were used to facilitate model building for low-resolution regions. The model so obtained was manually adjusted and rebuilt in COOT combined with iterative refinement with phenix.real_space_refine[66] imposing secondary structural restraints. All models were validated using MolProbity v4.5. It is to be noted that although we prepared a complex of CXCR3 in the presence of CXCL10, we could not observe any density for CXCL10, and therefore treated this structure as an Apo-state structure. Data collection, processing, and model refinement statistics are included in Supplementary Fig. 9. All figures in the manuscript were prepared using either Chimera-1.15 or ChimeraX-1.5 packages[70,71].

## Screening all CXCRs with VUF11418/VUF10661

To determine the specificity of VUF11418 and VUF10661, the two ligands were screened against the entire panel of CXC receptors in 3 assays: GloSensor Assay (to measure cAMP response), NanoBiT-based

G-protein dissociation assay and NanoBiT-based βarr1/2 recruitment assay. HEK293T cells were transiently transfected during splitting. Briefly, cells were trypsinized, pooled, and resuspended in incomplete media. This was followed by incubation of cells (1.2 million cells for each reaction) with transfection mix and subsequent seeding in 96-well plates at a density of 80,000 cells/well. For GloSensor assay, 1 μg of N-terminally FLAG-tagged receptor and 1 μg of F22 (Promega, Cat. no: E2301) was used to transfect HEK293T cells; for NanoBiT-based G-protein dissociation assay, 0.5 μg of N-terminally FLAG-tagged receptor, 1 μg of GoB tagged with LgBiT at its N-terminus, 1.5 μg of Gβ and 1.5 μg of Gγ tagged with SmBiT at its N-terminus were used to transfect HEK293T cells while for NanoBiT-based βarr1/2 recruitment assay, 1 μg of N-terminally FLAG-tagged receptor harboring a C-terminal SmBiT tag and 1 μg of either LgBiT-βarr1 or LgBiT-βarr2 (i.e., βarr1/2 harboring an N-terminal LgBiT) were used to transfect HEK293T cells.

Incomplete media was replaced with complete media after 6–8 h. The next day, media was replaced with 100 μL assay buffer (For Glo-Sensor assay: 20 mM HEPES pH 7.4, 1× Hank's Balanced Salt Solution/ HBSS and 0.5 mg/mL D-luciferin (GoldBio, Cat. no: LUCNA-1G); For NanoBiT-based assays: 5 mM HEPES pH 7.4, 1× HBSS, 0.01% BSA and 10 μM coelenterazine (GoldBio, Cat. no: CZ05)). The plates were first incubated at 37 °C for 1 h 30 min followed by an additional 30 min at room temperature.

For the GloSensor assay, basal luminescence was measured for 5 cycles using a multiwell plate reader (BMG Labtech). Since we measured Gi-mediated decrease in cytosolic cAMP levels, we added 5 μM forskolin to each well, to facilitate an increase in cAMP levels, and measured luminescence for 8 cycles. We then added the different ligands at the indicated final concentration and measured luminescence for 30 cycles.

For NanoBiT-based assays, basal luminescence was recorded for 3 cycles using a multiwell plate reader (BMG Labtech). Ligand was added at the indicated final concentrations and luminescence was recorded for 20 cycles. An average of the luminescence observed for cycles 5–9 was taken. The signal observed was normalized with respect to the luminescence observed at the lowest concentration of each ligand, treated as either 100% (for GloSensor assay) or 1 (for NanoBiT assay). Data was plotted and analyzed using GraphPad Prism 10 software.

### GloSensor assay to measure agonist-induced decrease in cytosolic cAMP

Agonist-induced decrease in cytosolic cAMP levels, as a readout of Gi-mediated second messenger signaling, was measured using GloSensor Assay, as previously described[52,53]. Briefly, HEK293T cells were transiently transfected with 3.5 μg of N-terminally FLAG-tagged CXCR3/ CXCR7 and 3.5 μg of F22 (Promega, Cat. no: E2301). 14–16 h post-transfection, the cells were washed with 1× PBS, trypsinized, and resuspended in assay buffer (20 mM HEPES pH 7.4, 1× Hank's Balanced Salt Solution/ HBSS and 0.5 mg/mL D-luciferin (GoldBio, Cat. no: LUCNA-1G)) and seeded in 96-well plates at a density of 100,000 cells/well. This was followed by an incubation of 1 h 30 min at 37 °C and another 30 min at room temperature. Data was collected, analyzed, and plotted as mentioned in the preceding section. For screening the various mutants of CXCR3, HEK293T cells were transiently transfected with the following mixture of DNA: 3.5 μg of F22 and either 3.5 μg of CXCR3$^{WT}$/ CXCR3$^{Y60A}$/ CXCR3$^{W109A}$/ CXCR3$^{F131A}$/ CXCR3$^{N132A}$/ CXCR3$^{F135A}$/ CXCR3$^{Q219A}$/ CXCR3$^{Y308A}$/ CXCR3$^{Y271A}$/ CXCR3$^{G307A}$ or 2 μg of CXCR3$^{S304A}$. All the receptors used harbored an N-terminal FLAG-tag. Data presented in Fig. 3a, b, Fig. 4a, Fig. 5a, Fig. 9d, and Fig. 10a indicate a % normalized response as measured using a decrease in forskolin-induced cAMP levels upon ligand stimulation.

### NanoBiT-based G-protein dissociation assay

Agonist-induced G-protein dissociation using a NanoBiT-based assay was measured as previously described[73]. For screening the various mutants of CXCR3, HEK293T cells were transiently transfected with the following mixture of DNA: 1 μg of GoB tagged with LgBiT at its N-terminus, 4 μg of Gβ, 4 μg of Gγ tagged with SmBiT at its N-terminus and 1 μg of either CXCR3$^{WT}$/ CXCR3$^{Y60A}$/ CXCR3$^{W109A}$/ CXCR3$^{F131A}$/ CXCR3$^{N132A}$/ CXCR3$^{F135A}$/ CXCR3$^{Q219A}$/ CXCR3$^{Y308A}$/ CXCR3$^{Y271A}$/ CXCR3$^{G307A}$/ CXCR3$^{S304A}$. All the receptors used harbored an N-terminal FLAG-tag. 14–16 h following transfection, the cells were washed with 1× PBS, trypsinized, and seeded in 96-well plates at a density of 100,000 cells/well in the presence of assay buffer (5 mM HEPES pH 7.4, 1× HBSS, 0.01% BSA and 10 μM coelenterazine (GoldBio, Cat. no: CZ05)). The plates were first incubated at 37 °C for 1 h 30 min followed by an additional 30 min at room temperature. Data was collected, analyzed, and plotted as mentioned in the preceding section. Data presented in Fig. 3a, b, Fig. 4a, Fig. 5a and Fig. 9d indicate a % normalized response as measured using a decrease in luminescence signal upon ligand stimulation as a readout of heterotrimer dissociation.

### NanoBiT-based βarr recruitment assay

To measure agonist-induced βarr1/2 recruitment downstream of CXCR3, we used a previously described NanoBiT-based assay[52,53]. In brief, for measuring βarr1/2 recruitment, HEK293T cells were transiently transfected with 3.5 μg of either CXCR3-SmBiT or CXCR7-SmBiT (i.e., receptor bearing an N-terminal FLAG-tag and a C-terminal SmBiT tag) and 3.5 μg of either LgBiT-βarr1 or LgBiT-βarr2 (i.e., βarr harboring an N-terminal LgBiT-tag). 14–16 h after transfection, the cells were washed with 1× PBS, trypsinized and seeded in 96-well plates at a density of 100,000 cells/well in the presence of assay buffer (5 mM HEPES pH 7.4, 1× HBSS, 0.01% BSA and 10 μM coelenterazine (GoldBio, Cat. no: CZ05)). The plates were first incubated at 37 °C for 1 h 30 min followed by an additional 30 min at room temperature. Data was collected, analyzed, and plotted as mentioned in the preceding section.

For screening the various mutants of CXCR3, βarr1 recruitment was measured in bystander mode. HEK293T cells were transiently transfected with the following mixture of DNA: 5 μg of Lg-CAAX, 2 μg of βarr1 bearing an N-terminal SmBiT tag and either 3 μg of CXCR3$^{WT}$/ CXCR3$^{Y60A}$/ CXCR3$^{W109A}$/ CXCR3$^{F131A}$/ CXCR3$^{N132A}$/ CXCR3$^{F135A}$/ CXCR3$^{Q219A}$/ CXCR3$^{Y308A}$/ CXCR3$^{Y271A}$/ CXCR3$^{G307A}$ or 2 μg of CXCR3$^{S304A}$. All the receptors used harbored an N-terminal FLAG-tag.

For screening the various mutants of CXCR7, βarr1 recruitment was measured in direct mode, wherein HEK293T cells were transiently transfected with the following mixture of DNA: 2 μg of βarr1 bearing an N-terminal LgBiT-tag and either 1.5 μg of CXCR7$^{WT}$/ CXCR7$^{Y51A}$/ CXCR7$^{W100A}$/ CXCR7$^{F124A}$/ CXCR7$^{S125A}$/ CXCR7$^{L128A}$/ CXCR7$^{S216A}$/ CXCR7$^{Q301A}$/ CXCR7$^{L305A}$ or 3 μg of CXCR7$^{Y268A}$/ CXCR7$^{S304A}$. The total amount of DNA was made up to 7 μg with pcDNA3.1. All the receptors used harbored an N-terminal FLAG-tag and a C-terminal SmBiT tag.

### Measuring βarr recruitment using the TANGO assay

To validate that the dual-agonism exhibited by VUF10661 and VUF11418 is not an experimental artifact, we measured βarr2 recruitment to CXCR7 using TANGO assay[74]. In brief, HTLA cells were transfected with 7 μg of CXCR7 harboring an N-terminal FLAG-tag and a C-terminal TEV protease cleavage site followed by the tTA transcription factor. 24 h post-transfection, cells were trypsinized and seeded in 96-well plates at a density of 100,000 cells/well in complete DMEM media. After another 24 h, complete media was replaced with incomplete media, and cells were stimulated with the indicated concentration of ligand for an additional 6 h at 37 °C. Following this, media in the wells was replaced with the assay buffer (20 mM HEPES pH 7.4, 1× Hank's Balanced Salt Solution/ HBSS and 0.5 mg/mL D-luciferin (GoldBio, Cat. no: LUCNA-1G)). Luminescence was recorded immediately in a microplate reader (BMG Labtech). The signal observed was normalized with respect to the luminescence observed at the lowest concentration of each ligand, treated as 1. Data was plotted and analyzed using GraphPad Prism 10 software.

## Receptor surface expression

Receptor surface expression was measured using whole-cell ELISA[75]. HEK293T cells expressing FLAG-tagged receptors were seeded in 24-well plates at a density of 0.2 million cells/well and allowed to adhere overnight. The next day, media was removed from the wells, and cells were washed once with 400 µL 1× TBS. Cells were fixed by incubating with 300 µL of 4% (w/v) paraformaldehyde/PFA for 20 min and excess PFA was removed by washing thrice with 400 µL 1× TBS. Wells were blocked with 200 µL 1% BSA prepared in 1× TBS for 1 h and then incubated with 1:10,000 anti-FLAG M2-HRP (Sigma-Aldrich, Cat. no: A8592) for another 1 h. Excess antibody was removed by washing thrice with 400 µL 1% BSA. Signal was developed by adding 200 µL of tetramethylbenzidine/TMB (Thermo Fisher Scientific, Cat. no: 34028). Once adequate color developed, the reaction was quenched by transferring 100 µL of the solution to a 96-well plate containing 100 µL of 1 M $H_2SO_4$. Absorbance was recorded at 450 nm using a multimode plate reader (Victor X4-Perkin-Elmer). In order to normalize the response observed across wells, cell density was quantified using Janus Green. Excess TMB solution was removed from the wells and the wells were washed once with 400 µL of 1× TBS. Thereafter, the wells were incubated with 200 µL of 0.2% (w/v) Janus Green for 15–20 min. Excess stain was removed by washing three times with distilled water and color was developed by adding 800 µL of 0.5 N HCl to each well. 200 µL of this colored solution was transferred to a 96-well plate and absorbance was recorded at 595 nm. Surface expression of the receptor was normalized by taking the ratio of the signal observed at 450 nm to the signal observed at 595 nm. For all cellular experiments, receptors were expressed at the cell surface at comparable levels (Supplementary Fig. 10).

## Reporting summary

Further information on research design is available in the Nature Portfolio Reporting Summary linked to this article.

## Data availability

All the data are included in the manuscript and any additional information required to reanalyze the data reported in this paper is available from the corresponding author upon request. The cryo-EM structures are deposited in Protein Data Bank (PDB) and Electron Microscopy Data Bank (EMDB) with accession numbers 8XXY and EMD-38765 for Apo-CXCR3-Go (Receptor-Ligand Focused), 8XXZ and EMD-38766 for Apo-CXCR3-Go (Full map), 8Y0H and EMD-38803 for VUF11418-CXCR3-Go (Receptor-Ligand Focused), 8Y0N and EMD-38809 for VUF11418-CXCR3-Go (Full map), 8XYI and EMD-38774 for VUF10661-CXCR3-Go (Receptor-Ligand Focused), 8XYK and EMD-38776 for VUF10661-CXCR3-Go (Full map). Source data are provided with this paper.

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

## Acknowledgements

Research in A.K.S.'s laboratory is supported by the Senior Fellowship of the DBT Wellcome Trust India Alliance (IA/S/20/1/504916) awarded to A.K.S., the Science and Engineering Research Board (SPR/2020/000408 and IPA/2020/000405), the Indian Council of Medical Research (F.NO.52/15/2020/BIO/BMS), and IIT Kanpur. A.K.S. is Sonu Agrawal Memorial Chair Professor. S.S. is funded by the Prime Minister's Research Fellowship (PMRF). This work was supported by grants from the JSPS KAKENHI, grant numbers 21H05037 (O.N.), 22K19371 and 22H02751 (W.S.), and 23KJ0491 (F.K.S.); the Kao Foundation for Arts and Sciences (W.S.); the Takeda Science Foundation (W.S.); the Lotte Foundation (W.S.); and the Platform Project for Supporting Drug Discovery and Life Science Research [Basis for Supporting Innovative Drug Discovery and Life Science Research (BINDS)]2 from the Japan Agency for Medical Research and Development (AMED), grant numbers JP22ama121012 (O.N.) and JP22ama121002 (support number 3272; O.N.). We thank Sayantan Saha, Samanwita Mohapatra, and Dr. Manish K. Yadav for their help in protein purification for the reconstitution of the complexes.

## Author contributions

S.S. and S.S.h. expressed and purified CXCR3, reconstituted the complexes for structural analysis, and carried out the functional assays together with A.D., S.M., D.T., and N.Z.; FKS prepared and screened the cryo-EM grids, collected and processed the cryo-EM data, and solved the structures with help from H.A., T.A.K., and Y.I.; M.G. and R.B. refined and analyzed the structures and prepared the figures with input from S.S. and N.R.; R.L. provided small molecule agonists of CXCR3; R.B., W.S., O.N. and A.K.S. supervised the overall study.

## Competing interests

The authors declare no competing interests.
