## [Transparent Peer Review file · Nature Communications]

Structural visualization of small molecule recognition by CXCR3 uncovers dual-agonism in the CXCR3-CXCR7 system

Corresponding Author: Professor Arun Shukla

Version 0:

Reviewer comments:

Reviewer #1

(Remarks to the Author)

Overall the manuscript is significantly improved. The comparisons to other CXCR3 and CXCR7 structures along with the site-directed mutagenesis provides important insights into signaling by these receptors. The removal of the phosphoproteomics from the manuscript is not a negative as it used different agonists and there were no firm conclusions from those studies.

Reviewer #3

(Remarks to the Author)

The authors have addressed all my comments. I recommend accepting this manuscript after correction of the following minor points.

1. In Extended Data Fig. 6, the residue numbering in the 'Total residues' category for miniG α appears inconsistent with the 'Resolved residues'. In the native G α sequence, Y354 is the last residue, consistent with the numbering of the resolved residues. However, T172-Y366 is listed as total range of residues in the expression construct. Could the authors clarify the specific elements included in their expression construct, particularly the positions of any affinity tags and protease cleavage sites?
2. In Fig. 3a,b, and 9d, the scale bars of G protein assays might cause confusion. Perhaps the authors can specify in the figure legends whether the cAMP response represents cAMP production or inhibition. Similarly, it would be helpful to indicate in the legends whether the GoB dissociation reflects an increase or decrease relative to the control.
3. In line 183, Y2716.51Ala should be changed to Y2716.51A.

1 **Response to reviewers**

2 **Reference:** NCOMMS-25-01705

3 **Title:** Structural visualization of small molecule recognition by CXCR3 uncovers dual-agonism
4 in the CXCR3-CXCR7 system

5 **Reviewer #1**

6 Overall, the manuscript is significantly improved. The comparisons to other CXCR3 and
7 CXCR7 structures along with the site-directed mutagenesis provides important insights into
8 signaling by these receptors. The removal of the phosphoproteomics from the manuscript is
9 not a negative as it used different agonists and there were no firm conclusions from those
10 studies.

11 We thank the reviewer for their positive comment and support.

12 **Reviewer #3**

13 The authors have addressed all my comments. I recommend accepting this manuscript after
14 correction of the following minor points.

15 We thank the reviewer for their positive comment and support. We have addressed the
16 remaining minor points in the revised manuscript as outlined below.

17 1. In Extended Data Fig. 6, the residue numbering in the 'Total residues' category for miniGao
18 appears inconsistent with the 'Resolved residues'. In the native Gao sequence, Y354 is the
19 last residue, consistent with the numbering of the resolved residues. However, T172-Y366 is
20 listed as total range of residues in the expression construct. Could the authors clarify the
21 specific elements included in their expression construct, particularly the positions of any affinity
22 tags and protease cleavage sites?

23 The construct used here contains an N-terminal hexa-histidine tag followed by a TEV protease
24 cleavage site, and a GGSGGSGG linker sequence replacing the His⁵⁷-Thr¹⁸² of the \$\alpha\$ -helical

25 domain. We have now mentioned this in the methods section of the revised manuscript (line
26 389-391, page 16) and also revised the residue numbering in Supplemental Data Fig. 6
27 accordingly.

28 2. In Fig. 3a,b, and 9d, the scale bars of G protein assays might cause confusion. Perhaps the
29 authors can specify in the figure legends whether the cAMP response represents cAMP
30 production or inhibition. Similarly, it would be helpful to indicate in the legends whether the
31 GoB dissociation reflects an increase or decrease relative to the control.

32 This is an excellent point, and indeed, we are presenting the normalized response. We have
33 now clarified this in the corresponding figure legends, and also mentioned it in the methods
34 section of the revise manuscript (line 585-587, page 24 and line 599-601, page 25).

35 3. In line 183, Y2716.51Ala should be changed to Y2716.51A.

36 We have made the corresponding change in the revised manuscript (line 183, page 8).